# PHYSICS-GUIDED MOTION LOSS FOR VIDEO GENERATION MODEL

## ABSTRACT

Current video diffusion models generate visually compelling content but often violate basic laws of physics, producing subtle artifacts like rubber-sheet deformations and inconsistent object motion. We introduce a frequency-domain physics prior that improves motion plausibility without modifying model architectures. Our method decomposes common rigid motions (translation, rotation, scaling) into lightweight spectral losses, requiring only 2.7% of frequency coefficients while preserving 97%+ of spectral energy. Applied to Open-Sora, MVDIT, and Hunyuan, our approach improves both motion accuracy and action recognition by $\sim 11\%$ on average on OpenVID-1M (relative), while maintaining visual quality. User studies show 74–83% preference for our physics-enhanced videos. It also reduces warping error by 22–37% (depending on the backbone) and improves temporal consistency scores. These results indicate that simple, global spectral cues are an effective drop-in regularizer for physically plausible motion in video diffusion.

## 1 INTRODUCTION

Diffusion-based video generation has recently achieved impressive frame quality. However, even flagship text–to–video diffusion systems still struggle with *physical motion*. Typical issues include rubber-like stretching, periodic flicker, and incorrect zooms or rotations. In short, frames can look good, but the motion often does not follow simple rules such as constant-velocity translation, rigid rotation, and uniform scaling.

Prior work to incorporate "physics" into video models mainly falls into four groups. (i) Flow/warping consistency helps reduce flicker but can break under large motion, occlusion, or brightness changes (Ho et al., 2020; Fleet & Jepson, 1990; Teed & Deng, 2020). (ii) Geometry-/3D-aware conditioning (e.g., depth or camera priors) improves rigidity but is domain-specific and computationally heavy in open settings (Harvey et al., 2022; Blattmann et al., 2023). (iii) Physics-inspired rules (e.g., constant-velocity penalties) help in narrow cases but do not scale well (Guen & Thome, 2020; Kataoka et al., 2020; Chen et al., 2023). (iv) Test-time refinements can smooth results but do not change the motion that the model actually learns.

Our idea is to regularize motion in the frequency domain, where basic physical motions leave simple, easy-to-detect patterns. Instead of matching pixels frame by frame, we look at global cues that summarize how energy moves over time. This view—formalized by a SIM(2) framework for translation, rotation, and scaling—brings three practical benefits: it is global (not pairwise), more tolerant to brightness or small rendering errors, and helps separate translation, rotation, and scale without hand-tuned rules (Adelson & Bergen, 1985; Bracewell, 1956; Simoncelli & Heeger, 1998).

We turn these ideas into a frequency loss that encourages generated videos to show frequency patterns consistent with basic physical motion. A simple adaptive weighting focuses on whatever motion pattern is most supported in the current clip while remaining sensitive to mixtures. The loss is computed on a truncated spectrum for efficiency and drops into standard diffusion training without changing the backbone.

Contributions: (1) A frequency-based theoretical framework that connects basic physical motion to simple frequency-domain patterns, offering global, robust cues that reduce common motion ambiguities.

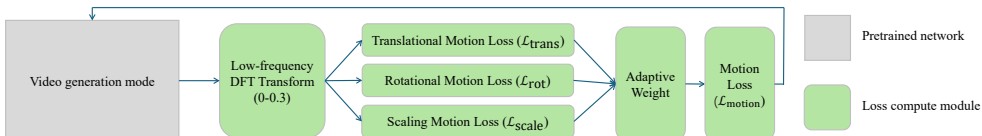

Figure 1: Our pipeline. From the generated video, we compute a low-pass 3D FFT, feed the spectral features to three motion losses (translation, rotation, scaling), and adaptively combine them into $\mathcal{L}_{\text{motion}}$ for training.

(2) A differentiable motion-aware regularizer with adaptive weighting that improves the learned motion behavior and handles mixed motion without hard classification.
(3) Experiments across multiple diffusion backbones show consistent gains in motion and temporal stability, visual quality and text–video alignment.

## 2 RELATED WORK

**Frequency-Domain Representation of Physical Motion.** The frequency representation of motion has deep roots: early studies modeled motion perception in frequency space (Watson & Ahumada, 1985), and phase consistency was shown to encode motion, inspiring optical-flow methods (Fleet & Jepson, 1990). Translational motion was characterized spectrally (Simoncelli & Heeger, 1998); rotations yield annular energy with discrete temporal peaks (Bracewell, 1956); scaling manifests as radial energy flow (Chen et al., 2010). Frequency analysis supports motion energy filtering (Adelson & Bergen, 1985) and multiband decompositions for motion layering and segmentation (Wang & Adelson, 1994). Recent work exploits spectral traits for motion classification (Sevilla-Lara et al., 2016) and improves interpolation via frequency cues (Xue et al., 2019). For evaluation, spectral metrics quantify spatiotemporal coherence of generated videos (Tesfaldet et al., 2018), though most use frequency only at *evaluation* time rather than in training.

**Physics-Constrained Video Generation.** Integrating physics into generative modeling is emerging: video prediction with physical consistency (Guen & Thome, 2020) and action generation under constraints (Kataoka et al., 2020). These methods, however, typically target specific constraints and lack a general framework for physical motion. MotionCraft (Savant Aira et al., 2024) imposes physics-based motion at inference via simulated flow warping of image-diffusion noise, while our method regularizes motion during video-diffusion training in the frequency domain.

**Deep Learning-Based Video Generation Models.** GAN-based methods separate static/dynamic components (Vondrick et al., 2016), decouple temporal and image generators (Saito et al., 2017), or disentangle motion/content (Tulyakov et al., 2018). Autoregressive models include flow-based VideoFlow (Kumar et al., 2020) and transformer tokenization (Weissenborn et al., 2020), but face temporal and computational limits. Latent-space approaches leverage VAEs and decomposed latents for high-resolution, temporally coherent generation (Yan et al., 2019; Villegas et al., 2019).

**Diffusion Models for Video Generation.** Diffusion has advanced video quality via joint spatiotemporal denoising (Ho et al., 2020), spatiotemporal U-Nets (Harvey et al., 2022), probabilistic factorization (Yang et al., 2023), and latent-space diffusion for efficiency (Blattmann et al., 2023). Conditional generation scales further with text and multimodal conditioning (Singer et al., 2022; Wu et al., 2023) and motion modules (Guo et al., 2023). Despite rapid progress, including Sora (Brooks et al., 2024) and Step-Video-T2V (Ma et al., 2025), maintaining physically plausible motion remains challenging.

## 3 PHYSICS-GUIDED MOTION-AWARE LOSS FUNCTION

State-of-the-art video diffusion models are typically trained with data-driven objectives plus optional optical-flow or temporal-smoothness terms (Ho et al., 2020; Harvey et al., 2022; Blattmann et al., 2023; Guo et al., 2023). While these reduce flicker, they do not explicitly encode basic physical motion (constant-velocity translation, rigid rotation, uniform scaling), so artifacts that *look* like

motion—rubber-sheet deformations, periodic flicker, scale/zoom glitches—persist even in simple scenes (Brooks et al., 2024; Ma et al., 2025).

We ground our remedy in frequency space. Within a unified SIM(2) spectral framework (translations, rotations, uniform scalings) (Sharma & Duits, 2015)[1] spectral geometry, physically plausible motions exhibit simple slice-wise signatures: (i) *rotation*: energy aligns with tilted lines $\omega_t + m\Omega = 0$ in $(m, \omega_t)$ and concentrates annularly; (ii) *scaling*: radial–temporal gradients align and the radial spectral centroid shows a clear monotone trend; (iii) *translation*: energy lies near a plane in $(\omega_x, \omega_y, \omega_t)$.

Guided by these diagnostics, we add a small, differentiable frequency-domain regularizer on $\hat{x}_0$ composed of three *slice-consistent* losses (translation, rotation, scaling) with adaptive weighting. Because full SIM(2) hyperplane suffers from (1) cross-slice interference and energy double counting (mixing $\{\omega_x, \omega_y\}$ with $m, \nu$), (2) ill-conditioning under mixed motions when $|m|$ or $|\nu|$ excitation is weak, and (3) coupled weighting that obscures which failure mode is being corrected.

## 3.1 FREQUENCY-DOMAIN CHARACTERISTICS OF PHYSICAL MOTION

Different types of physical motion exhibit unique and distinguishable features in the frequency domain, providing a theoretical foundation for our loss function design. We first briefly outline these features and then discuss how we design loss functions based on them.

**SIM(2) spectral manifold.** Consider a short temporal window where the dominant motion is well-approximated by a similarity transform (Hartley & Zisserman, 2004) (translation $v = (v_x, v_y)$, in-plane rotation with angular velocity $\Omega$, and isotropic scaling rate $\alpha = \dot{\sigma}$ in log-radius). Let $\widehat{V}(\omega_x, \omega_y, \omega_t)$ be the FFT-based (Oppenheim, 1999) spatiotemporal spectrum. Passing to polar coordinates $(\rho, \theta)$ in the spatial frequency plane and expanding along the angular and log-radial axes yields harmonic indices $m \in \mathbb{Z}$ and $\nu \in \mathbb{Z}$. Then the ideal SIM(2) motion concentrates spectral energy ($E(\omega_x, \omega_y, \omega_t) = |\widehat{V}(\omega_x, \omega_y, \omega_t)|^2$) on a single hyperplane in $(\omega_x, \omega_y, m, \nu, \omega_t)$:

$$\omega_t + v_x \omega_x + v_y \omega_y + \Omega m + \alpha \nu + b_0 = 0, \qquad (3.1)$$

$b_0$ is a regression intercept that absorbs residual phase offsets and discretization bias; see App A.1 for the windowing convention. Three classical facts are recovered as special cases: (i) **translation**: $\omega_t + v_x \omega_x + v_y \omega_y = 0$ (a plane in $(\omega_x, \omega_y, \omega_t)$); (ii) **rotation**: $\omega_t + \Omega m = 0$ (tilted lines in $(m, \omega_t)$, since the $m$-th angular harmonic acquires a temporal factor $e^{-im\Omega t}$); (iii) **scaling**: $\omega_t + \alpha \nu = 0$ (tilted lines in $(\nu, \omega_t)$). These equal the translation/rotation/scaling slices of the SIM(2) spectral hyperplane; derivations are in App. A.2, and practical choices (polar/log-radius resampling, energy/observability gating, robust regression) are in App. A.6. Figure 2 visualizes these spectral slices on synthetic SIM(2) sequences.

**Multi-object and mixed motions.** Although equation 3.1 is derived for a single SIM(2) motion in a short window, by linearity of the 3D FFT, multiple SIM(2) motions give rise to an approximately additive superposition of their spectral patterns. This behavior is visible in the bottom rows of Fig. 2, where two independently translating objects and a moving and shrinking object generate multiple structures in the translation, rotation, and scaling slices.

**Energy-weighted unified residual .** From the observed spectrum(the 3D FFT of the current $T \times H \times W$ video window; energy $E = |\widehat{V}|^2$) we build samples $(\phi_i, b_i, w_i)$ with design vector

$$\phi_i = [\omega_x, \omega_y, m, \nu, 1], \qquad b_i = -\omega_t,$$

and energy/observability weight $w_i$ (details in App. A.1 energy gate, low-$m/\nu$ suppression, Huber/Charbonnier robustification). We estimate $\boldsymbol{\theta} = [v_x, v_y, \Omega, \alpha, b_0]^\top$ by weighted ridge regression

$$\hat{\boldsymbol{\theta}} = \arg \min_{\boldsymbol{\theta}} \sum_i w_i (\boldsymbol{\phi}_i \boldsymbol{\theta} - b_i)^2 + \lambda \|\boldsymbol{\theta}\|_2^2.$$

and define the unified residual $\mathcal{L}_{\text{uni}} = \frac{\sum_i w_i (\phi_i \hat{\theta} - b_i)^2}{\sum_i w_i}$. If the window follows a SIM(2) motion, $\mathcal{L}_{\text{uni}}$ tends to 0 as $T \to \infty$. For general motions and finite windows, $\mathcal{L}_{\text{uni}}$ *upper-bounds and controls*

---

[1]The 2D similarity group (translations, rotations, and uniform scalings): it acts on image-plane points as $x' = s R(\theta) x + t$ with $t \in \mathbb{R}^2$, $R(\theta) \in \text{SO}(2)$, and $s > 0$ (4 DoF). Shear and anisotropic scaling are excluded.

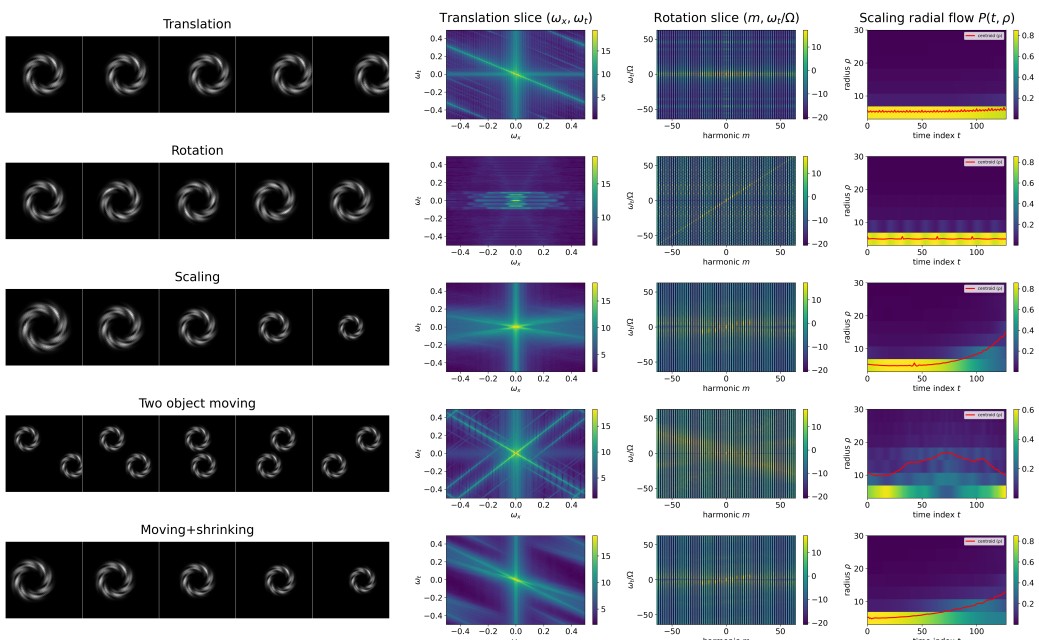

Figure 2: Visualization of SIM(2) spectral signatures on synthetic sequences. Rows show different motions; columns show sample frames, the translation slice, the rotation slice, and the scaling radial-flow map with radial centroid(red). Pure SIM(2) motions (top rows) produce the expected simple structures in each slice, while the multi-object and mixed-motion rows show superpositions that remain visually separable.

the off-hyperplane energy fraction under the band tolerance $\Delta$ (cf. Lemma B.2 and Lemma B.3). *Sketch.* Combine the classical translation plane, rotational angular-harmonic identity $\delta(\omega_t + \Omega m)$, and log-radial shift identity $\delta(\omega_t + \alpha\nu)$ under Parseval; see App A.2 and A.3 for details.

**Differentiable WLS.** We estimate motion parameters with an energy-weighted ridge least-squares fit (differentiable solve; $\lambda = 10^{-3}$, jitter ($\varepsilon = 10^{-8}$), FP32; pseudo-inverse fallback).

### 3.1.1 FREQUENCY-DOMAIN CHARACTERISTICS OF TRANSLATIONAL MOTION

For translational motion with constant velocity $(v_x, v_y)$, its spatiotemporal representation is:

$$V(x, y, t) = V_0(x - v_x t, y - v_y t).$$ (3.2)

In the frequency domain, this motion concentrates energy on an *affine* plane:

$$\omega_t + \omega_x v_x + \omega_y v_y + b_0 = 0,$$ (3.3)

where $b_0$ absorbs windowing/phase conventions (in the ideal unwindowed case $b_0=0$). This is translation slice of the SIM(2) model (cf. Eq. equation 3.1) (Adelson & Bergen, 1985; Bracewell, 1956; Simoncelli & Heeger, 1998). The normal vector $(v_x, v_y, 1)$ is directly related to the motion velocity. In our synthetic example, the panel in the first row, second column of Figure 2 show that the energy in $E_{x,t}(\omega_x, \omega_t)$ (obtained by averaging over $\omega_y$) concentrates along an approximately straight ridge, consistent with the affine plane in equation 3.3.

### 3.1.2 FREQUENCY-DOMAIN CHARACTERISTICS OF ROTATIONAL MOTION

Under in-plane rotation with angular velocity $\Omega$,

$$V(r, \theta, t) = V_0(r, \theta - \Omega t).$$

The spatiotemporal frequency analysis implies two signatures (see App. A.2): (i) *annular* spatial energy concentration (from the Bessel-type radial response of angular harmonics), and (ii) *tilted*

*lines* in the $(m, \omega_t)$ plane given by

$$\omega_t + m\Omega = 0,$$

i.e., the $m$-th angular harmonic carries a temporal tone at $m\Omega$. Under finite temporal windows these appear as narrow bands (cf. Lemma B.3). This is precisely the rotational slice of the unified SIM(2) plane in Eq. equation 3.1 (Bracewell, 1956; Adelson & Bergen, 1985; Simoncelli & Heeger, 1998; Fleet & Jepson, 1990). In the Figure 2 (third column in each row), we normalize temporal frequency by the ground-truth angular velocity $\Omega$, so that the ideal relation $\omega_t + m\Omega = 0$ becomes straight rays $\omega_t/\Omega \approx -m$ passing through the origin in the $(m, \omega_t/\Omega)$ plane.

### 3.1.3 FREQUENCY-DOMAIN CHARACTERISTICS OF SCALING MOTION

A spatial scaling by a factor $s > 0$ obeys the Fourier-scaling law in 2D:

$$\mathcal{F}\{V_0(x/s, y/s)\}(\omega_x, \omega_y) = s^2 \widehat{V}_0(s\,\omega_x, \, s\,\omega_y),$$

so scaling induces a radial reallocation of spectral energy (Brigham, 1988; Oppenheim, 1999; Mallat, 1999). For a time-varying scale $s(t) = e^{\sigma(t)}$, passing to log-polar coordinates $(\rho, \theta) \mapsto (\xi = \log \rho, \theta)$ turns scaling into a translation $\xi \mapsto \xi - \sigma(t)$; consequently, after angular/log-radial expansions the energy in $(\nu, \omega_t)$ concentrates along the line

$$\omega_t + \alpha\,\nu = 0, \qquad \alpha = \dot{\sigma}(t),$$

which is precisely the scaling slice of our unified SIM(2) model (cf. Eq. equation 3.1) (Lindeberg, 1993; Reddy & Chatterji, 1996). Operationally, this appears as a "radial energy flow": zoom-in (increasing $s$) shifts energy to lower spatial frequencies, and zoom-out to higher ones. The rightmost column of Figure 2 shows the corresponding radial-flow map $P(t, \rho)$ and the radial centroid $\rho_c(t)$ (red); for the pure scaling row, the centroid follows a clear monotone trend, matching the expected radial energy flow induced.

For training-time measurements we adopt two simple proxies. (i) A radial–temporal gradient alignment score $C_{\text{flow}}$ (dot product of normalized $\nabla_\rho E$ and $\nabla_t E$) captures the presence and direction of radial flow. (ii) The temporal trend of the radial spectral centroid $\rho_c(t)$ provides a scale-rate proxy (derivations and bounds in App. §A.4–A.5).

### 3.2 COMPUTABLE BOUNDS FOR WINDOW/INTERPOLATION

For a chosen temporal window $h$ (Hann), the RHS of Lemma B.3 gives a closed-form/lookup bound on $\varepsilon_{\text{win}}(\Delta)$ as a function of $(T, \Delta)$. The polar/log-radius bilinear interpolation error can be upper-bounded via local Lipschitz constants of the spectrum times grid spacings; in practice we calibrate $\varepsilon_{\text{interp}}$ on synthetic data by sweeping spectral gradients and reporting the worst-case relative energy error carried outside the target band.

### 3.3 TRANSLATIONAL MOTION LOSS

For constant-velocity translation $(v_x, v_y)$, the spectral support concentrates near a plane in $(\omega_x, \omega_y, \omega_t)$ as 3.3. We estimate the parameters via energy-weighted least squares (WLS). Let $A_i = (\omega_{x,i}, \omega_{y,i}, 1)$, $b_i = -\omega_{t,i}$, $\beta_{\text{tr}} = [\,v_x, v_y, b_0\,]^\top$. With weights $\mathbf{W}_{ii} \geq 0$ (energy/observability gating; App. A.6), the ridge-WLS estimator and the normalized residual are

$$\hat{\beta}_{\text{tr}} = \arg\min_{\beta_{\text{tr}}} \sum_i \mathbf{W}_{ii} \left(A_i \beta_{\text{tr}} - b_i\right)^2 + \lambda \|\beta_{\text{tr}}\|_2^2, \qquad \mathcal{L}_{\text{trans}} = \frac{\sum_i \mathbf{W}_{ii} \left(A_i \hat{\beta}_{\text{tr}} - b_i\right)^2}{\sum_i \mathbf{W}_{ii}}.$$

This plane model is the translation slice of our SIM(2) framework and is stable under short temporal windows; non-constant velocities appear as bandwidth broadening along $\omega_t$ and are naturally penalized by the residual (see App. A.6 for windowing and tolerance).

### 3.4 ROTATIONAL MOTION LOSS

Rotational motion exhibits two complementary spectral signatures: (i) annular spatial concentration; (ii) energy alignment along tilted lines $\omega_t + \Omega m = 0$ in the $(m, \omega_t)$ plane. We therefore adopt the

ring-concentration term and tilted-line energy ratio that matches the rotational slice of the unified SIM(2) model.

We estimate the angular velocity with an energy-weighted least squares:

$$\Omega^\star \;=\; -\,\frac{\sum_\rho \sum_{m\neq 0} \sum_{\omega_t} |\widetilde{C}_m(\rho,\omega_t)|^2\,\omega_t m}{\sum_\rho \sum_{m\neq 0} \sum_{\omega_t} |\widetilde{C}_m(\rho,\omega_t)|^2\,m^2}.$$

The tilted-line energy ratio and ring concentration are

$$C_{\text{rot}} \;=\; \frac{E_{\text{line}}}{E_{\text{all}}}, \quad E_{\text{line}} = \sum_{\rho,\,m\neq 0,\,|\omega_t+m\Omega^\star|\leq\Delta} |\widetilde{C}_m|^2, \qquad C_{\text{ring}} = 1 - \frac{H_{\text{ring}}}{\log N_r}.$$

($\widetilde{C}_m(\rho,\omega_t)$ denotes the time-DFT of the angular harmonic $C_m(\rho,t)$; $H_{\text{ring}}$ is the entropy of energy over $N_r$ concentric rings; $\Delta$ is one temporal-frequency bin in the condition $|\omega_t + m\Omega^\star| \leq \Delta$.)

$$\mathcal{L}_{\text{rot}} \;=\; 1 \;-\; \frac{C_{\text{ring}} + C_{\text{rot}}}{2}.$$

Under mild narrow-band and window assumptions, $\mathcal{L}_{\text{rot}}$ upper-bounds the unified SIM(2) rotational-slice residual up to window/interpolation terms (proof in App. A.5). Compared with prior "temporal-peak" heuristics, our line-ratio with energy-weighted $\Omega^\star$ suppresses non-rotational periodicities while remaining differentiable. Implementation details (polar resampling, $m{=}0$ exclusion, $\Delta$, and weighting) are in App. D.2 and App. A.6.

## 3.5 Scaling Motion Loss

Scaling leaves a distinctive spectral signature: *radial energy flow* and *tilted lines* in the $(\nu,\omega_t)$ plane obeying $\omega_t + \alpha\nu = 0$, i.e., the scaling slice of the unified SIM(2) hyperplane (cf. Eq. equation 3.1). To keep the main text lightweight yet theory-linked, we adopt two robust proxies and state their consistency with the SIM(2) slice; the closed-form $\alpha^\star$ and a tilted-line ratio $C_{\text{scale}}$ are given in App. A.4–A.5.

Let $E_k(t)$ be the ring energy on the $k$-th annulus (App. D.2). Define $E_k(t)$ and their unit fields:

$$\nabla_\rho E_{k,t} = E_{k+1,t} - E_{k,t}, \quad \nabla_t E_{k,t} = E_{k,t+1} - E_{k,t},$$

$$\hat{\nabla}_\rho E = \frac{\nabla_\rho E}{\sqrt{\sum_{k,t}|\nabla_\rho E_{k,t}|^2 + \varepsilon}}, \qquad \hat{\nabla}_t E = \frac{\nabla_t E}{\sqrt{\sum_{k,t}|\nabla_t E_{k,t}|^2 + \varepsilon}}.$$

The (direction-agnostic) alignment score is

$$C_{\text{flow}} = \left|\sum_{k,t} \hat{\nabla}_\rho E_{k,t} \cdot \hat{\nabla}_t E_{k,t}\right|. \quad \rho_c(t) = \frac{\sum_k k\,E_k(t)}{\sum_k E_k(t) + \varepsilon_{\text{stab}}},$$

Let $\rho_c(t)$ denote the radial spectral centroid and define a bounded trend strength via correlation:

$$S_{\text{trend}} = \left|\text{corr}\big(\rho_c, t\big)\right| = \frac{\left|\text{cov}(\rho_c, t)\right|}{\sqrt{\text{var}(\rho_c)\,\text{var}(t) + \varepsilon}}.$$

**Scaling loss.** ($E_k(t)$ is the spectral energy on the $k$-th concentric ring in the spatial-frequency plane (optionally normalized per $t$); $\varepsilon, \varepsilon_{\text{stab}} > 0$ are small positive numerical-stability constants.)

$$\mathcal{L}_{\text{scale}} \;=\; 1 - \frac{C_{\text{flow}} + S_{\text{trend}}}{2}.$$

Under a log-polar narrow-band shift model $E(i,t) \approx A(i - u(t))$ with $u'(t) = \alpha$ and mild window/interp errors, $\mathcal{L}_{\text{scale}}$ upper-bounds the unified SIM(2) scaling-slice residual up to constants (proof and the closed-form $\alpha^\star$ with the $(\nu,\omega_t)$ tilted-line ratio $C_{\text{scale}}$ in App. A.4–A.5). For very short windows ($T{<}3$) we default both proxies to 0.5.

Table 1: Quantitative evaluation on the OpenVID-1M dataset. Best results are in **bold**.

(a) Open-Sora

| Metric | Baseline | Ours |
|---|---|---|
| VQA_A ↑ | 65.15 | **69.2** |
| VQA_T ↑ | 59.57 | **69.71** |
| SD Score ↑ | 68.24 | **68.42** |
| CLIP Temporal Score ↑ | 99.80 | **99.85** |
| Warping Error ↓ | 0.0089 | **0.0056** |
| Temporal Consistency ↑ | 61.45 | **63.82** |
| Action Recognition Score ↑ | 60.77 | **69.71** |
| Motion Accuracy Score ↑ | 44.00 | **49.00** |
| Flow Score ↑ | 1.15 | **1.18** |
| Text-Video Alignment ↑ | 54.02 | **61.05** |
| BLIP-BLEU ↑ | 23.73 | **24.52** |

(b) MVDIT

| Metric | Baseline | Ours |
|---|---|---|
| VQA_A ↑ | 66.65 | **69.3** |
| VQA_T ↑ | 63.96 | **70.24** |
| SD Score ↑ | 68.31 | **68.78** |
| CLIP Temporal Score ↑ | 99.83 | **99.89** |
| Warping Error ↓ | 0.0080 | **0.0062** |
| Temporal Consistency ↑ | 62.21 | **64.73** |
| Action Recognition Score ↑ | 62.34 | **69.70** |
| Motion Accuracy Score ↑ | 44.0 | **51.0** |
| Flow Score ↑ | 1.01 | **1.22** |
| Text-Video Alignment ↑ | 61.04 | **63.62** |
| BLIP-BLEU ↑ | 24.14 | **24.76** |

## 3.6 ADAPTIVE WEIGHTING AND COMPOSITE LOSS

To accommodate different videos that may contain multiple motion patterns, we introduce an adaptive weighting mechanism based on a temperature parameter $\tau$: $w_i^{\text{type}} = \frac{\exp(-\mathcal{L}_i/\tau)}{\sum_{j \in \mathcal{M}} \exp(-\mathcal{L}_j/\tau)}$. The theoretical foundation of this mechanism is grounded in the maximum-entropy principle from information theory. This gives higher weights to motion types with lower loss values (i.e., better conforming to specific motion patterns). When $\tau$ approaches 0, the weight distribution becomes more "winner-takes-all," highlighting the optimal motion type; higher $\tau$ values produce a smoother weight distribution, suitable for mixed motion. The final composite loss is the weighted sum of the motion losses:

$$\mathcal{L}_{\text{motion}} = \sum_{i \in \mathcal{M}} w_i^{\text{type}} \cdot \mathcal{L}_i. \tag{3.4}$$

This design not only enables the model to automatically identify the dominant motion type but also maintains sensitivity to mixed motion. Compared to traditional hard classification or single motion assumption methods, this approach better handles complex motion scenarios in the real world.

## 4 EXPERIMENTAL RESULTS

To evaluate the effectiveness of our proposed frequency domain-based approach for physical motion enhancement in video generation, we conducted extensive experiments on state-of-the-art video diffusion models. We demonstrate that our method improves motion quality and physical plausibility while maintaining or enhancing visual quality and semantic alignment.

### 4.1 EXPERIMENTAL SETUP

**Low-pass truncation.** We keep a per-dimension low-pass *cube* with fraction $\varrho=0.3$ along $(\omega_t, \omega_x, \omega_y)$, so only $\varrho^3 = 2.7\%$ of spectral coefficients are processed (thus $\approx 97.3\%$ coefficient-level FLOPs reduction). Under a radial power-law spectrum $E(\boldsymbol{\omega}) \propto \|\boldsymbol{\omega}\|^{-2\kappa}$ ($\kappa \approx 1.8$), the ball-retained energy fraction admits a closed form:

$$\eta_{\text{ball}}(\varrho) = \frac{(\varrho R)^{3-2\kappa} - \varepsilon^{3-2\kappa}}{R^{3-2\kappa} - \varepsilon^{3-2\kappa}}. \tag{4.1}$$

Our cube satisfies tight geometric bounds:

$$\eta_{\text{ball}}(\varrho) \leq \eta_{\text{cube}}(\varrho) \leq \eta_{\text{ball}}\big(\min\{1, \sqrt{3}\varrho\}\big). \tag{4.2}$$

At $\varrho=0.3$ and typical video sizes, this yields $\eta_{\text{cube}}(0.3) \in [0.97, 0.987]$. All derivations and a numerical sanity check are in App C.

**Datasets and Models.** We conducted our experiments using the OpenVID-1M dataset, a large-scale open-domain video dataset containing diverse motion patterns. For baseline models, we selected three video diffusion models: Open-Sora (Zheng et al., 2024), MVDIT (Nan et al., 2024), and Hunyuan (Kong et al., 2025) which represent different architectural approaches to video diffusion.

Table 2: Evaluation results of finetune Hunyuan model (Kong et al., 2025).

| Metric | Hunyuan-Baseline | LoRA, Base loss | LoRA, Flow Loss | LoRA, Ours Loss |
|---|---|---|---|---|
| VQA_A ↑ | 73.85 | 73.84 | 73.92 | **74.79** |
| VQA_T ↑ | 85.14 | 85.16 | 85.36 | **87.37** |
| SD Score ↑ | 68.32 | 68.42 | 68.37 | **69.83** |
| CLIP Temporal Score ↑ | 99.91 | 99.90 | 99.92 | **99.95** |
| Warping Error ↓ | 0.0024 | 0.0022 | 0.0022 | **0.0016** |
| Temporal Consistency ↑ | 63.65 | 63.70 | 64.23 | **66.03** |
| Action Recognition Score ↑ | 68.93 | 69.02 | 68.78 | **73.15** |
| Motion Accuracy Score ↑ | 56.00 | 56.0 | 56.0 | **59.0** |
| Flow Score ↑ | 1.39 | 1.39 | 1.45 | **1.46** |
| Text-Video Alignment ↑ | 59.60 | 60.62 | 60.33 | **65.34** |
| BLIP-BLEU ↑ | 24.41 | 24.42 | 24.60 | **25.23** |

Table 3: Evaluation of simple and complex motion.

| Metric | Baseline-complex | Ours-complex | Baseline-simple | Ours-simple |
|---|---|---|---|---|
| SD Score ↑ | 70.20 | 71.34 | 67.82 | 69.42 |
| CLIP Temporal Score ↑ | 99.92 | 99.96 | 99.90 | 99.95 |
| Warping Error ↓ | 0.0020 | 0.0011 | 0.0025 | 0.0018 |
| Flow Score ↑ | 1.39 | 1.46 | 1.38 | 1.46 |
| BLIP-BLEU ↑ | 26.41 | 26.92 | 23.87 | 24.77 |

**Implementation details.** We fine-tune each backbone for four epochs using a cosine-annealed LR initialized at $2 \times 10^{-5}$ on $4 \times$ NVIDIA A100 GPUs. At every diffusion step $t$ we reconstruct $\hat{x}_0$, evaluate the physics-informed frequency loss on $\hat{x}_0$, and add it to the standard denoising objective. The weighted least-squares block is fully differentiable (no detaching of $\hat{\theta}$), runs in FP32 with autocast off, and uses ridge $\lambda = 10^{-3}$ and $\varepsilon = 10^{-8}$ (rest in BF16); see App. A.6–C for windows, polar/log resampling, gating and stability details.

**Evaluation protocol.** Following EvalCrafter (Liu et al., 2023) and the OpenVID-1M setting, we report four categories of metrics to enable fair comparison on this benchmark: *visual quality* (VQA_A, VQA_T, SD-Score), *temporal coherence* (CLIP-Temporal, Warping Error, Temporal Consistency), *motion quality* (Action Recognition, Motion Accuracy, Flow), and *text alignment* (Text–Video Alignment, BLIP-BLEU). We use the official EvalCrafter implementation and evaluation protocol for OpenVID-1M to ensure comparability with prior work.

## 4.2 QUANTITATIVE RESULTS

Table 1a presents our quantitative evaluation results on the Open-Sora model. Our approach consistently outperforms the baseline across most metrics. Notably, we observe substantial improvements in motion-related metrics, with a 7% absolute improvement in Action Recognition Score and a 5% improvement in Motion Accuracy Score. This confirms that our frequency domain-based approach effectively enhances the physical plausibility of motion patterns in generated videos.

Table 1b shows similar improvements on the MVDIT model, indicating that our approach generalizes well across different video diffusion architectures. The consistent performance gains across both models validate the effectiveness of our physics-informed frequency domain approach.

Importantly, our method achieves these motion quality improvements without sacrificing visual quality or semantic alignment. We observe modest improvements in these dimensions as well, suggesting that enhancing physical motion plausibility can positively influence overall video generation quality.

We further evaluate on a recent Physics Generation Benchmark (Meng et al., 2025) that isolates mechanics, optics, thermal, and material phenomena. Our method improves the average physics score from 0.44 to 0.52 over Open-Sora (Table 4), suggesting better adherence to basic physical laws.

**Comparison on Hunyuan (Kong et al., 2025) (LoRA).** For the *Hunyuan* model, we adopt LoRA (PEFT) due to its size, inserting low-rank adapters into attention and time-related linear layers while

Table 4: Evaluation on the Physics Generation Benchmark.(Meng et al., 2025)

| Metric | Mechanics↑ | Optics↑ | Thermal↑ | Material↑ | Average↑ |
|---|---|---|---|---|---|
| Open-Sora | 0.37 | 0.44 | 0.37 | 0.37 | 0.44 |
| Ours | 0.45 | 0.55 | 0.54 | 0.55 | 0.52 |

**Ours**                                    **Baseline**

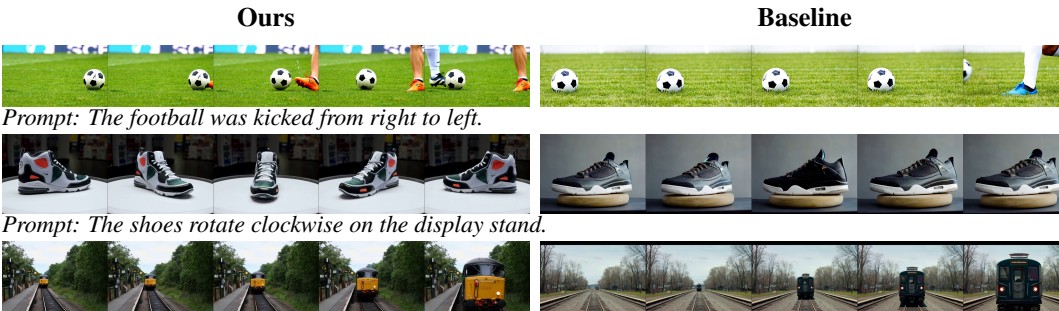

*Prompt: The football was kicked from right to left.*

*Prompt: The shoes rotate clockwise on the display stand.*

*Prompt: The train moves from far away to nearby, and the camera is fixed to create a zooming effect.*

Figure 3: Comparison between ours and the baseline (Hunyuan) under translation, rotation, and scaling.

freezing all other weights. To ensure fairness, the baseline and "+Ours" use identical LoRA placement, rank, and training schedule. Under this fixed adapter budget, our frequency-domain physical loss yields consistent gains in *Motion Quality* and maintains (or slightly improves) *Visual Quality* and *Text Alignment*. (see Table 2).

**Flow-based Temporal Consistency (Strong Baseline)** While optical flow has long been used to impose temporal coherence via warping-based photometric losses in video generation/processing (e.g., video style transfer and video-to-video translation), most contemporary text-to-video diffusion systems do *not* include an explicit flow-matching loss during training; they typically rely on architectural inductive biases, cross-frame conditioning, or inference-time alignment. For completeness and a strong baseline, , we implement a standard flow-based consistency loss. Given adjacent frames $x_t, x_{t+1}$, we estimate forward/backward optical flow $\mathbf{F}_{t\to t+1}, \mathbf{F}_{t+1\to t}$ with RAFT (Teed & Deng, 2020). Let $\mathcal{W}$ be bilinear warping and $\mathbf{M}, \mathbf{M}'$ be visibility masks. Our loss is a masked reprojection error with robust Charbonnier/$\ell_1$ penalty plus a standard flow smoothness term:

$$\mathcal{L}_{\text{flow}} = \left\| \mathbf{M} \odot \left( x_{t+1} - \mathcal{W}(x_t, \mathbf{F}_{t\to t+1}) \right) \right\|_1 + \left\| \mathbf{M}' \odot \left( x_t - \mathcal{W}(x_{t+1}, \mathbf{F}_{t+1\to t}) \right) \right\|_1 + \lambda \mathcal{L}_{\text{smooth}}(\mathbf{F}).$$
(4.3)

This provides a temporal-consistency regularizer that directly constrains cross-frame correspondences, is sensitive to jitter/stretching artifacts, needs no backbone changes, and adds controllable overhead (RAFT inference only). As shown in Table 2, Our model outperforms the strong baseline.

**Stratified evaluation by motion complexity.** To isolate the effect of our physics-guided motion loss, we stratify test prompts into *simple* vs. *complex* motion using an LLM (GPT-5) following our rubric: a prompt is *simple* when the dominant motion is well-approximated by a single rigid transform (translation, rotation, scaling); otherwise it is *complex*. The complete split is provided in the Supplementary. We then recompute metrics for the *baseline* and *baseline + ours* within each stratum. As shown Table 3, We stratify by motion complexity and observe consistent improvements over the baseline in both the simple and complex subsets. On average, gains are larger on the simple subset.

**Ablation study** In our ablation study on the Open-Sora (Zheng et al., 2024) model (Table 5), we find that each motion-specific loss term contributes uniquely to overall performance. Omitting the translation loss leads to noticeable drops in both visual quality and basic motion fidelity, while removing the rotation loss disproportionately impairs semantic alignment and cyclical motion consistency. Excluding the scaling loss produces the smallest overall degradation but still measurably worsens zoom-related coherence and motion accuracy. Importantly, none of the individual removals recovers the balanced improvements achieved by the full composite loss, demonstrating that translation, ro-

Table 5: Ablation study of individual motion loss components on the OpenVID-1M dataset using the Open-Sora model. We report the full loss and variants where the translation loss, rotation loss, or scaling motion loss is removed. All models were fine-tuned for four epochs.

| Metric | Full Loss | w/o Translation Loss | w/o Rotation Loss | w/o Scaling Loss |
|---|---|---|---|---|
| VQA_A ↑ | **69.20** | 66.15 | 66.95 | 67.10 |
| VQA_T ↑ | **69.71** | 64.80 | 65.00 | 65.40 |
| SD Score ↑ | **68.42** | 67.85 | 68.10 | 68.20 |
| CLIP Temporal Score ↑ | **99.85** | 99.70 | 99.75 | 99.80 |
| Warping Error ↓ | **0.0056** | 0.0078 | 0.0070 | 0.0068 |
| Temporal Consistency ↑ | **63.82** | 61.20 | 62.10 | 62.18 |
| Action Recognition Score ↑ | **69.71** | 66.20 | 66.50 | 67.00 |
| Motion Accuracy Score ↑ | **49.00** | 44.00 | 45.00 | 44.00 |
| Flow Score ↑ | **1.18** | 1.15 | 1.16 | 1.16 |
| Text-Video Alignment ↑ | **61.05** | 57.60 | 58.00 | 58.20 |
| BLIP-BLEU ↑ | **24.52** | 23.80 | 23.92 | 23.43 |

tation, and scaling constraints act synergistically to yield the best trade-off across visual, temporal, and motion-quality metrics.

**User Study and Analysis** We adopted the Two-Alternative Forced-Choice (2AFC) protocol to evaluate whether our design improves different backbone models. For each trial, participants were shown two videos side-by-side—one from a vanilla baseline model, the other from the same model augmented with ours and asked to choose which looked better overall. We integrated our approach into three representative state-of-the-art video generation models and pooled all comparisons together. A total of 106 participants each completed 15 randomly ordered trials, with left–right placement of baseline vs. augmented outputs independently randomized to eliminate positional bias. Viewers were instructed to consider both visual quality (sharpness, color fidelity) and motion naturalness (motion smoothness, coherence) in the first question.

As shown in Figure 6, our augmented models were preferred over their vanilla counterparts across every backbone, with overall preference rates ranging from 74.2 % to 82.7 % depending on the architecture. The user study interface as shown in Figure 4. These results demonstrate that our method not only boost performance on a single model, but generalize effectively across multiple distinct video generation pipelines.

### 4.3 QUALITATIVE ANALYSIS

Fig. 3 contrasts our method with the Hunyuan baseline on three canonical motions. (i) Translation. For a sphere instructed to move left-to-right, the baseline remains nearly static for $\sim$!80% of the sequence, whereas our method produces smooth, monotonic displacement. (ii) Clockwise rotation. The baseline reverses direction mid-sequence (left then right), breaking temporal consistency; our method maintains a single, persistent clockwise rotation. (iii) Forward motion with scale. For a train advancing toward the camera, the baseline shows abrupt appearance of the train with jerky forward motion, while ours yields smooth central approach with the expected increase in scale (see supplementary video). These qualitative observations are consistent with our quantitative results and indicate that our frequency-domain formulation better preserves directionality, phase, and scale dynamics across motion types. Perhaps unexpectedly, we observed failure cases under simple SIM(2) prompts; examples are included in the supplementary.

## 5 DISCUSSION AND CONCLUSION

**Limitations and future work.** Our framework currently addresses only basic rigid motions (translation, rotation, scaling) and does not yet model more complex or non-rigid dynamics such as elastic deformations or articulated movements. Extending beyond these fundamental patterns is left for future work.

**Conclusion.** We have introduced a unified, frequency-domain approach for enforcing physical motion plausibility in video diffusion models. Our method consistently improves action recognition, optical flow quality, and motion accuracy—while preserving visual fidelity and semantic alignment.

**Reproducibility statement.** We provide theory, implementation choices, and evaluation protocols to facilitate independent reproduction. The frequency-domain formulation and motion-aware losses (translation/rotation/scaling) are specified in the main *Physics-Guided Motion-Aware Loss Function* section, with core identities (e.g., equation 3.1, equation 3.3) and a SIM(2) slice-consistent view. Complete derivations of the SIM(2) spectral manifold and the unified gold-standard residual, together with surrogate–residual upper-bound relations, are given in App. A.2, App. A.3, App. A.5, and the sufficient statistics in App. A.4. Practical implementation details—including polar/log-radius resampling, energy/observability gating, robust/ridge WLS solvers, and numeric precision—are centralized in App. A.6, with low-pass truncation modeling and retained-energy bounds in App. C (see equation 4.1, equation 4.2); the translation and rotation components are further detailed in App. D.1 and App. D.2. Training setup and hardware, together with a concise reproducibility checklist, are summarized in the main *Experimental Setup* and App. A.8. Datasets and evaluation follow the OpenVID-1M/EvalCrafter protocol described in *Datasets and Models* and *Evaluation protocol*; quantitative results and comparisons are reported in Tab. 1, Tab. 2, Tab. 3, with ablations in Tab. 5, and the strong flow-consistency baseline is specified in Sec. 4.2. Qualitative visualizations and the processing pipeline appear in Fig. 3, Fig. 5, and Fig. 1; the 2AFC user-study design and outcomes are provided in *User Study and Analysis*, Fig. 6, and Fig. 4. We also report stratified evaluation by motion complexity in the main text (complete splits in the Supplementary).

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

# A APPENDIX

## A.1 NOTATION AND TRANSFORMS

We denote the input video window by $V \in \mathbb{R}^{T \times H \times W}$ (one channel for simplicity; RGB is handled channel-wise with energies summed). We work in the complex spatiotemporal Fourier domain. Let

$$\widehat{V}(\omega_x, \omega_y, \omega_t) = \mathrm{DFT}_t\Big\{ h[t] \cdot \mathrm{DFT}_{x,y}\big\{ V(t,\cdot,\cdot) - \tfrac{1}{2} \big\} \Big\},$$

be the separable 2D spatial DFT per frame followed by a 1D temporal DFT (with an optional Hann window $h[t]$); frequencies are indexed by $(\omega_x, \omega_y, \omega_t)$. On the spatial frequency plane we adopt polar coordinates $(\rho, \theta)$ with $\rho = \sqrt{\omega_x^2 + \omega_y^2}$ and $\theta = \mathrm{atan2}(\omega_y, \omega_x)$. Angular harmonics are obtained via a DFT over $\theta$, yielding $C_m(\rho, \omega_t)$ indexed by $m \in \mathbb{Z}$; log-radial harmonics are obtained by re-sampling $\rho$ on a logarithmic grid $\xi = \log \rho$ and applying a 1D DFT to obtain $D_\nu(\omega_t)$ indexed by $\nu \in \mathbb{Z}$. We compute energies as $E(\cdot) = \big|\widehat{V}(\cdot)\big|^2$.

## A.2 UNIFIED SIM(2) SPECTRAL MANIFOLD: DERIVATION

We consider a short temporal window where the dominant motion is well-approximated by a similarity transform: translation $v = (v_x, v_y)$, in-plane rotation with angular velocity $\Omega$, and isotropic scaling rate $\alpha = \dot{\sigma}$ (with $s(t) = e^{\sigma(t)}$).

**Translation.** For $V(x, y, t) = V_0(x - v_x t, y - v_y t)$, the DFT analysis gives the classical plane constraint

$$\omega_t + v_x \omega_x + v_y \omega_y = 0. \tag{A.1}$$

**Rotation.** Write $V(r, \theta, t) = V_0(r, \theta - \Omega t)$. Expanding $V_0$ in angular harmonics and transforming in $t$ yields

$$\widehat{V}(\rho, \theta, \omega_t) = \sum_{m \in \mathbb{Z}} e^{im\theta} \mathcal{B}_m(\rho)\, \delta(\omega_t + m\Omega), \tag{A.2}$$

where $\mathcal{B}_n$ involves Bessel kernels (annular concentration). Hence energy concentrates along *tilted lines* $\omega_t + \Omega m = 0$ in the $(m, \omega_t)$ plane.

**Scaling.** Let $V(x, y, t) = V_0\big(\frac{x}{s(t)}, \frac{y}{s(t)}\big)$ with $s(t) = e^{\sigma(t)}$. In spatial frequency, scaling is a dilation; in log-radius $\xi = \log \rho$, it becomes a shift $\xi \mapsto \xi - \sigma(t)$. Therefore the $(\nu, \omega_t)$ spectrum concentrates on

$$\omega_t + \alpha\, \nu = 0, \qquad \alpha = \dot{\sigma}. \tag{A.3}$$

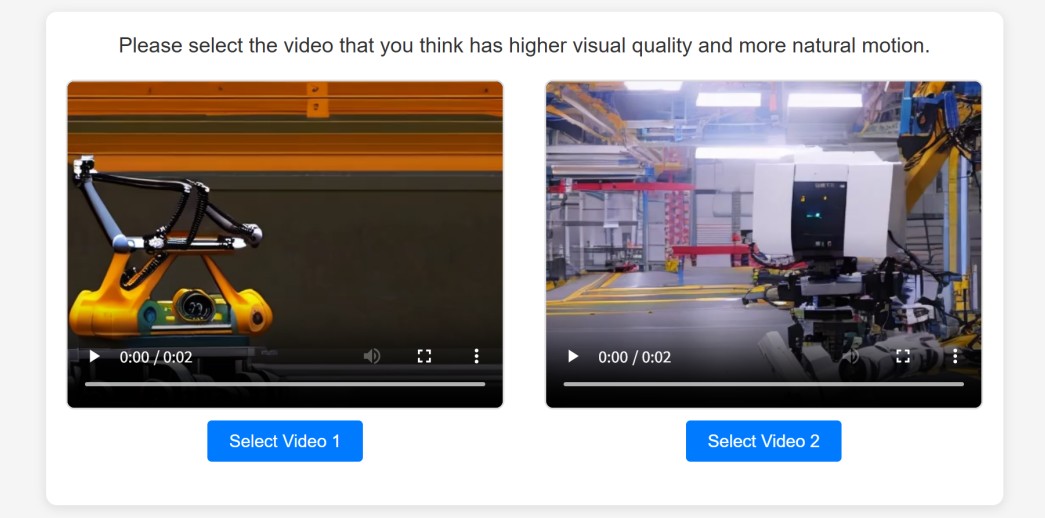

Figure 4: User study interface design. Our evaluation employs a two-alternative forced choice (2AFC) protocol with two types of questions. The first type (shown) asks participants to select the video with higher visual quality and more natural motion from a pair of generated videos displayed side-by-side. The second type evaluates text-video alignment by asking participants to choose which video better matches the given text prompt. Video positions are randomized to eliminate positional bias.

**Unified hyperplane.** Collecting the three facts above, an ideal SIM(2) motion concentrates spectral energy on a single hyperplane in $(\omega_x, \omega_y, m, \nu, \omega_t)$:

$$\omega_t + v_x\omega_x + v_y\omega_y + \Omega\, m + \alpha\, \nu + b_0 = 0, \tag{A.4}$$

where $b_0$ absorbs constant phase/windowing terms. Eq. equation A.1 and the rotational/scaling tilted lines are recovered by setting the other coefficients to zero.

### A.3    GOLD-STANDARD RESIDUAL AND BASIC PROPERTIES

From the observed spectrum we build weighted samples $(\phi_i, b_i, w_i)$ with

$$\phi_i = \begin{bmatrix} \omega_x, & \omega_y, & m, & \nu, & 1 \end{bmatrix}, \qquad b_i = -\omega_t, \qquad w_i \geq 0,$$

and estimate $\boldsymbol{\theta} = [\, v_x, v_y, \Omega, \alpha, b_0 \,]^\top$ via weighted ridge regression:

$$\hat{\theta} = \arg\min_\theta \sum_i w_i\, (\phi_i\theta - b_i)^2 + \lambda\|\theta\|_2^2. \tag{A.5}$$

The unified *gold-standard residual* is

$$\mathcal{L}_{\mathrm{uni}} = \frac{\sum_i w_i\, (\phi_i\hat{\theta} - b_i)^2}{\sum_i w_i}. \tag{A.6}$$

**Proposition 1 (Zero-residual for ideal SIM(2)).** If the video window follows an exact SIM(2) motion with constant parameters, then $\mathcal{L}_{\mathrm{uni}} = 0$ up to boundary/windowing terms. *Sketch.* The spectral mass is supported on the hyperplane equation A.4; any least-squares estimate lying on that plane gives zero orthogonal projection residual. Parseval's identity transfers the spatial transformations to spectral constraints; see standard derivations for translation planes, rotational angular-harmonic $\delta(\omega_t + m\Omega)$ lines, and log-radius temporal shifts.

**Proposition 2 (Consistency under noise).** Assume i.i.d. additive spectral noise with finite second moment, weights bounded and bounded away from zero on a set of positive measure, and sufficient excitation in $(\omega_x, \omega_y, m, \nu)$. Then $\hat{\theta} \to \theta^\star$ and $\mathcal{L}_{\mathrm{uni}} \to \sigma^2$ (noise floor) as the number of samples increases. Weighted ridge ensures a bounded condition number.

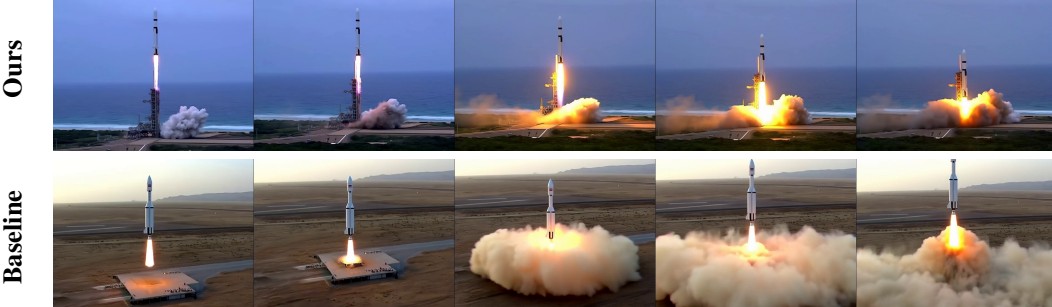

*Prompt: A rocket performs a controlled vertical landing onto a coastal pad.*

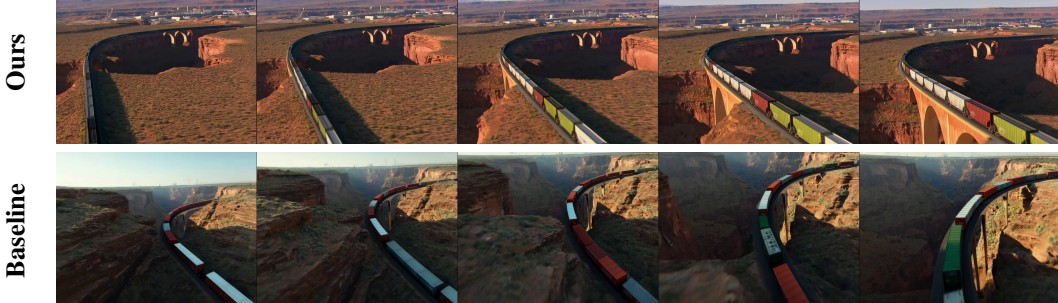

*Prompt: A freight train arcs through a canyon.*

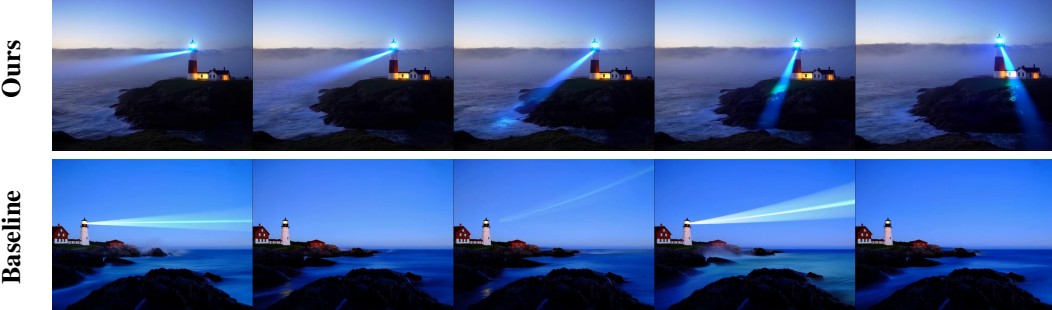

*Prompt:A lighthouse's rotating beacon sweeps; projected light cones translate and scale over rocks and waves.*

Figure 5: Additional comparisons with baseline (Hunyuan). In Figure 1, the baseline's rocket lands and then takes off again. In Figure 2, the train car colors change abruptly, and additional train cars appear out of nowhere. In Figure 3, the baseline's lighthouse light is inconsistent and doesn't generate according to the prompt instructions. However, our motion is all coherent.

**Observability.** The parameters are locally identifiable when the design matrix has full column rank in the energy-weighted sense, which requires non-degenerate support in $\{\omega_x, \omega_y\}$ for translation, nontrivial $|m|$ for rotation, and nontrivial $|\nu|$ for scaling.

### A.4 FROM THEORY TO COMPUTABLES: ROTATION AND SCALING SUFFICIENT STATISTICS

**Rotation.** Define the angular DFT and its temporal DFT:

$$C_m(\rho, t) = \frac{1}{2\pi} \int_0^{2\pi} \widehat{V}(\rho, \theta, t) e^{-im\theta} \, d\theta, \quad \widetilde{C}_m(\rho, \omega_t) = \text{DFT}_t\{C_m(\rho, t)\}.$$

An energy-weighted least-squares estimate for $\Omega$ has the closed-form

$$\Omega^\star = -\frac{\sum_{\rho, m, \omega_t} |\widetilde{C}_m|^2 \, \omega_t m}{\sum_{\rho, m, \omega_t} |\widetilde{C}_m|^2 \, m^2}. \tag{A.7}$$

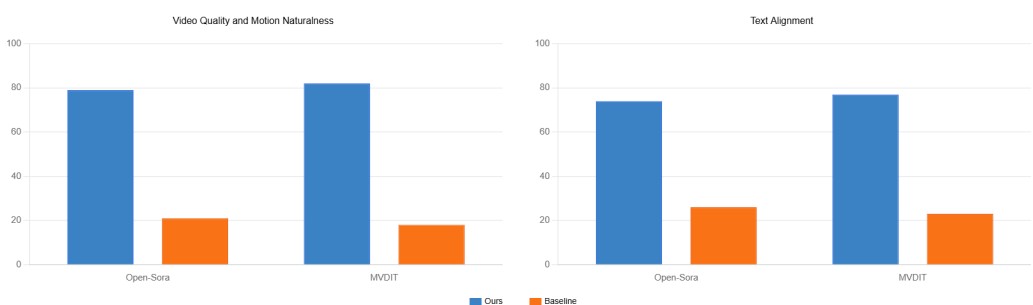

Figure 6: User study results. We compare our method to two baselines (Open-Sora and MVDIT) using a two-alternative forced choice protocol. For each baseline, we report the percentage of user votes in our favor (blue) and in favor of the baseline (orange). Our method was consistently preferred by users in both video quality/motion naturalness evaluation and text-prompt alignment assessment, achieving preference rates of 79.4% and 74.2% against Open-Sora, and 82.7% and 77.9% against MVDIT, respectively.

The *tilted-line energy ratio*

$$E_{\text{line}} = \sum_{\rho} \sum_{m \neq 0} \sum_{|\omega_t + m\Omega^\star| \leq \Delta} |\widetilde{C}_m|^2, \quad E_{\text{all}} = \sum_{\rho} \sum_{m \neq 0} \sum_{\omega_t} |\widetilde{C}_m|^2, \quad C_{\text{rot}} = \frac{E_{\text{line}}}{E_{\text{all}}} \tag{A.8}$$

directly measures concentration along $\omega_t + \Omega m = 0$. Combined with the annular concentration $C_{\text{ring}}$, our rotation loss is $\mathcal{L}_{\text{rot}} = 1 - \frac{1}{2}(C_{\text{ring}} + C_{\text{rot}})$.

**Scaling.** After log-radius re-sampling $\xi = \log \rho$ with a 1D-DFT along $\xi$, we obtain $D_\nu(t)$ and its temporal DFT $\widetilde{D}_\nu(\omega_t)$. The analogous estimate and tilted-line ratio are

$$\alpha^\star = -\frac{\sum_{\nu, \omega_t} |\widetilde{D}_\nu|^2 \, \omega_t \nu}{\sum_{\nu, \omega_t} |\widetilde{D}_\nu|^2 \, \nu^2}, \qquad C_{\text{scale}} = \frac{\sum_{|\omega_t + \alpha^\star \nu| \leq \Delta} |\widetilde{D}_\nu|^2}{\sum_{\omega_t, \nu} |\widetilde{D}_\nu|^2}. \tag{A.9}$$

In the main paper we keep two simpler, robust proxies: radial-flow consistency $C_{\text{flow}}$ and centroid trend $S_{\text{trend}}$. In §A.5 we show they upper-bound the unified residual on the scaling slice.

### A.5 Upper-Bound Relations to the Gold-Standard Residual

Let $\mathcal{L}_{\text{uni}}^{(\text{rot})}$ denote equation A.6 restricted to the rotational slice (i.e., translation/scaling features clamped). Under an energy narrow-band assumption (dominant annulus/ring) and mild noise, there exist constants $a, b > 0$ such that

$$1 - \tfrac{1}{2}(C_{\text{ring}} + C_{\text{rot}}) \ \leq \ a \cdot \mathcal{L}_{\text{uni}}^{(\text{rot})} + b \cdot \varepsilon_{\text{NB}}. \tag{A.10}$$

A similar bound holds for scaling with $C_{\text{flow}}$ and $S_{\text{trend}}$:

$$1 - \tfrac{1}{2}(C_{\text{flow}} + S_{\text{trend}}) \ \leq \ c_1 \cdot \mathcal{L}_{\text{uni}}^{(\text{scale})} + c_2 \cdot \varepsilon_{\text{NB}}. \tag{A.11}$$

*Sketch.* The tilted-line ratios are sufficient statistics for concentration in the $(m, \omega_t)$ or $(\nu, \omega_t)$ plane; Cauchy–Schwarz yields that the off-line energy controls the orthogonal projection residual to the corresponding slice of the unified plane. The ring/flow terms compensate for spatial localization (annular narrow-bandness), with $\varepsilon_{\text{NB}}$ capturing deviation from narrow-band energy.

For translation, the weighted plane-fitting residual used in the main paper is precisely a computable proxy for the projection residual onto Eq. equation A.1, and hence upper-bounds the unified residual on the translation slice.

### A.6 Implementation Details

**Polar interpolation.** We pre-compute a look-up table (LUT) to map polar bins $(\rho_k, \theta_\ell)$ to Cartesian indices $(\omega_x, \omega_y)$ and use bilinear interpolation on the spectral grid. We apply a Hann window in time before the temporal DFT to reduce leakage. Typical choices: $N_r \in \{16, 20\}$ rings (linear in $\rho$) and $M \in \{16, 24\}$ angular bins; for scaling, log-radius sampling uses $N_\xi \in \{16, 24\}$ bins.

**Observability and energy gates.** We use an energy gate $g_E = \sigma\big(f(\frac{E}{E_{\max}} - \tau_E)\big)$ with $\tau_E \in [0.1, 0.2]$ and $f \in [6, 10]$, and an observability gate for rotation $g_{\text{obs}}(m) = \frac{m^2}{m^2 + \lambda}$ with $\lambda = 1$. The final weight is $w_i = g_E \cdot g_{\text{obs}}$ (and analogous for scaling with $\nu$).

**Robust regression and regularization.** The unified regression is solved with ridge $\lambda = 10^{-3}$. In high-noise settings we replace squared loss by Huber/Charbonnier; the closed forms equation A.7, equation A.9 are retained for initialization, followed by one Gauss–Newton step. .

**Low-pass truncation.** We retain a low-frequency cube of fraction $\varrho = 0.3$ per dimension (time and two spatial axes), keeping $\varrho^3 = 2.7\%$ coefficients. The retained-energy bounds for cube vs. ball follow the derivation in the main paper; typical videos yield 97%–98.7% retained energy.

### A.7 COMPUTATIONAL COST

In our settings, wall-clock overhead is typically 10–20% of the backbone forward pass. ==Our method does not modify the backbone or sampler; inference compute is unchanged by design.==

### A.8 REPRODUCIBILITY CHECKLIST

- Hardware: NVIDIA A100 $\times$ 4; mixed precision BF16.
- Training: cosine LR schedule, initial LR $2\times10^{-5}$, window $T = 12$–$16$, loss weights as in the main paper.
- Spectral settings: rings $N_r{=}20$, angles $M{=}24$, log-radius bins $N_\xi{=}24$, tolerance $\Delta{=}1$ freq. bin, $\varepsilon{=}10^{-8}$.

## B RIGOROUS RELATIONS BETWEEN UNIFIED SPECTRAL RESIDUAL AND SURROGATE LOSSES

### B.1 NOTATION, SETTING, AND ASSUMPTIONS

**Spectral samples and design.** For each discrete spectral sample $i$, define

$$\phi_i = \big[\omega_x,\ \omega_y,\ m,\ \nu,\ 1\big], \qquad b_i = -\omega_t,$$

with energy $E_i \geq 0$, weight $w_i = g_i E_i$, and

$$\Phi = \begin{bmatrix} \phi_1^\top \\ \cdots \\ \phi_N^\top \end{bmatrix}, \quad b = \begin{bmatrix} b_1 \\ \cdots \\ b_N \end{bmatrix}, \quad W = \text{diag}(w_1, \ldots, w_N).$$

**Unified (weighted ridge) regression and residual.**

$$\hat{\theta}_\lambda = \arg\min_{\theta \in \mathbb{R}^5} \|W^{1/2}(\Phi\theta - b)\|_2^2 + \lambda\|\theta\|_2^2, \qquad \mathcal{L}_{\text{uni},\lambda} = \frac{\|W^{1/2}(\Phi\hat{\theta}_\lambda - b)\|_2^2}{\text{tr}(W)}.$$

For $\lambda = 0$ write $\hat{\theta}$, $\mathcal{L}_{\text{uni}}$. For rotational/scaling/translational *slices*, restrict $\phi_i$ to the corresponding subspace to obtain $\mathcal{L}_{\text{uni},\lambda}^{(\text{rot})}, \mathcal{L}_{\text{uni},\lambda}^{(\text{scale})}, \mathcal{L}_{\text{uni},\lambda}^{(\text{trans})}$.

**Surrogate losses (as implemented).**

- Translation translation: $\mathcal{L}_{\text{trans}} = \dfrac{\sum_i w_i e_i^2}{\sum_i w_i}$ with $e_i = \omega_t - q(\omega_x, \omega_y)$ for linear $q$.

- Rotation:
$$\mathcal{L}_{\text{rot}} = 1 - \tfrac{1}{2}\big(C_{\text{ring}} + C_{\text{rot}}\big),$$
where $C_{\text{rot}}$ is a tilted-line energy ratio on $(m, \omega_t)$ and $C_{\text{ring}}$ is the annular concentration.

- Scaling:
$$\mathcal{L}_{\text{scale}} = 1 - \tfrac{1}{2}\big(C_{\text{flow}} + S_{\text{trend}}\big).$$

**Time windowing and interpolation.** Let $h[t]$ be a temporal window (e.g., Hann) of length $T$; its DFT is $\hat{h}(\omega_t)$. Temporal windowing induces convolution and *leakage*; denote the relative out-of-band energy by $\varepsilon_{\text{win}}(\Delta)$ (defined later). Polar/log-radius resampling via bilinear interpolation induces an error $\varepsilon_{\text{interp}}$.

**Weights and gates.** Weights are $w_i = g_i E_i$ with $g_i = g_E(i) \cdot g_{\text{obs}}(i)$. Assume $g_E(i) \in [g_{\min}, 1]$ (energy gate) and $g_{\text{obs}}(i) \in [\tilde{g}_{\min}, \tilde{g}_{\max}]$, hence

$$g_i \in [\underline{g}, \overline{g}], \qquad \underline{g} := g_{\min} \tilde{g}_{\min} > 0, \ \ \overline{g} := \tilde{g}_{\max} < \infty.$$

**Narrow-band assumption (rotation/scaling).** At each time, at least $(1-\varepsilon)$ of the spatial spectral energy concentrates in a single annulus; $\varepsilon \in [0, 1)$.

**Observability.** Assume $\lambda_{\min}(\Phi^\top W \Phi) > 0$ (or $\lambda_{\min}(\Phi^\top W \Phi) + \lambda > 0$ with ridge) and the same for sliced subspaces.

## B.2 EXACTNESS FOR IDEAL SIM(2)

**Theorem B.1** (Asymptotic zero-residual). *Under a single SIM(2) motion, in the idealized continuous-time, infinite-window, and noise-free setting (no windowing or interpolation), the unified residual and slice surrogates tend to zero:*

$$\lim_{T \to \infty, \ \Delta \to 0^+} \left( \mathcal{L}_{\text{uni}}, \ \mathcal{L}_{\text{uni}}^{(\text{rot})}, \ \mathcal{L}_{\text{uni}}^{(\text{scale})}, \ \mathcal{L}_{\text{uni}}^{(\text{trans})}, \ \mathcal{L}_{\text{rot}}, \ \mathcal{L}_{\text{scale}}, \ \mathcal{L}_{\text{trans}} \right) = 0.$$

*With finite windows and discrete sampling, these losses are lower bounded by window/leakage and interpolation terms characterized in Lemma B.3.*

*Sketch.* Classical spectral support: translation on the plane $\omega_t + v_x \omega_x + v_y \omega_y = 0$; rotation on the line $\omega_t + \Omega m = 0$; scaling on the line $\omega_t + \alpha \nu = 0$. Least-squares projection to the true support yields zero residual; line-energy ratios equal 1; the annular entropy is 0. □

## B.3 BAND-CAPTURE AND WINDOW LEAKAGE

**Lemma B.2** (Weighted band-capture). *Let $e_i$ be algebraic distances to a target line/plane, and define*

$$E_{\text{in}}(\Delta) = \sum_{|e_i| \le \Delta} E_i, \qquad E_{\text{all}} = \sum_i E_i.$$

*If $w_i = g_i E_i$ with $g_i \in [\underline{g}, \overline{g}]$, then*

$$1 - \frac{E_{\text{in}}(\Delta)}{E_{\text{all}}} \ \le \ \frac{\overline{g}}{\underline{g}} \cdot \frac{1}{\Delta^2} \cdot \frac{\sum_i w_i e_i^2}{\sum_i w_i}. \tag{B.1}$$

*Proof.* Apply Chebyshev's inequality to the energy-weighted measure $p_i = E_i / E_{\text{all}}$: $\sum_{|e_i| > \Delta} p_i \le \Delta^{-2} \sum_i p_i e_i^2$. Substitute $p_i = \frac{w_i}{g_i} / \sum_j \frac{w_j}{g_j}$ and bound $g_i$ by $[\underline{g}, \overline{g}]$. □

**Lemma B.3** (Window leakage). *Let $\hat{h}(\omega_t)$ be the DFT of the temporal window $h[t]$. After convolution with $|H|^2$ in $\omega_t$, the relative out-of-band energy outside $\pm\Delta$ satisfies*

$$\varepsilon_{\text{win}}(\Delta) \ \le \ \frac{\sum_{|\omega_t| > \Delta} |\hat{h}(\omega_t)|^2}{\sum_{\omega_t} |\hat{h}(\omega_t)|^2}. \tag{B.2}$$

*For Hann/Blackman, the RHS decreases monotonically in $T$ and $\Delta$.*

## B.4 ROTATION: SURROGATE UPPER-BOUNDS UNIFIED SLICE RESIDUAL

**Rotation statistics.** The tilted-line ratio on $(m, \omega_t)$: $C_{\mathrm{rot}} = \frac{E_{\mathrm{in}}(\Delta)}{E_{\mathrm{all}}}$ with $e_i = \omega_t + m\Omega^\star$, and $\Omega^\star$ given by energy-weighted LS:

$$\Omega^\star = -\frac{\sum |\widetilde{C}_m(\rho, \omega_t)|^2 \, \omega_t m}{\sum |\widetilde{C}_m(\rho, \omega_t)|^2 \, m^2}.$$

Annular concentration: $C_{\mathrm{ring}} = 1 - \frac{H}{\log N_r}$ where $H$ is the entropy of the ring energy distribution.

**Lemma B.4** (Annulus entropy bound). *If at least $(1 - \varepsilon)$ of the energy lies in one ring, then*

$$1 - C_{\mathrm{ring}} = \frac{H}{\log N_r} \leq \frac{h(\varepsilon) + \varepsilon \log(N_r - 1)}{\log N_r}, \tag{B.3}$$

*with $h(\varepsilon) = -\varepsilon \log \varepsilon - (1 - \varepsilon) \log(1 - \varepsilon)$.*

**Theorem B.5** (Rotation surrogate $\leq$ unified slice residual). *For $\Delta \geq 1$ (one temporal-frequency bin at least), with window/interp errors $\varepsilon_{\mathrm{win}}(\Delta)$, $\varepsilon_{\mathrm{interp}}$ and radial narrow-bandness $\varepsilon$, one has*

$$\boxed{\mathcal{L}_{\mathrm{rot}} \leq \frac{\overline{g}}{2\underline{g}} \cdot \frac{1}{\Delta^2} \mathcal{L}_{\mathrm{uni},\lambda=0}^{(\mathrm{rot})} + \frac{h(\varepsilon) + \varepsilon \log(N_r - 1)}{2\log N_r} + \varepsilon_{\mathrm{win}}(\Delta) + \varepsilon_{\mathrm{interp}}.} \tag{B.4}$$

*Proof.* By Lemma B.2 with $e_i = \omega_t + m\Omega^\star$ and Lemma B.3, $1 - C_{\mathrm{rot}} \leq \frac{\overline{g}}{\underline{g}}\Delta^{-2}\mathcal{L}_{\mathrm{uni}}^{(\mathrm{rot})} + \varepsilon_{\mathrm{win}} + \varepsilon_{\mathrm{interp}}$. By Lemma B.4, $1 - C_{\mathrm{ring}} \leq \frac{h(\varepsilon) + \varepsilon \log(N_r - 1)}{\log N_r}$. Average the two bounds. $\square$

**Remark (ridge).** If a ridge version is used to estimate $\Omega^\star$, Theorem B.11 below implies $\mathcal{L}_{\mathrm{uni},\lambda}^{(\mathrm{rot})} \leq \mathcal{L}_{\mathrm{uni},0}^{(\mathrm{rot})} + \frac{\lambda}{\mathrm{tr}(W)}\|\theta_{\mathrm{LS}}\|_2^2$; absorb the $O(\lambda)$ term on the RHS of equation B.4.

## B.5 TRANSLATION

**Theorem B.6** (Band-in ratio vs. surrogate residual). *For any $\Delta \geq 1$, defining band-in energy around the fitted plane by $E_{\mathrm{in}}(\Delta)$, one has*

$$1 - \frac{E_{\mathrm{in}}(\Delta)}{E_{\mathrm{all}}} \leq \frac{\overline{g}}{\underline{g}} \cdot \frac{1}{\Delta^2} \mathcal{L}_{\mathrm{trans}} + \varepsilon_{\mathrm{win}}(\Delta). \tag{B.5}$$

*Here $e_i = \omega_{t,i} - q(\omega_{x,i}, \omega_{y,i})$ with the linear model $q(\omega_x, \omega_y) = \gamma_1 \omega_x + \gamma_2 \omega_y + \gamma_3$.*

*Proof.* Lemma B.2 with $e_i = \omega_t - q(\omega_x, \omega_y)$, plus window broadening by Lemma B.3. $\square$

**Theorem B.7** (Equivalence to unified slice residual). *Let $S$ be the* linear *feature subspace for the weighted inner product $\langle u, v \rangle = \sum_i w_i u_i v_i$. There exist $c_1, c_2 > 0$ (depending on the weighted Gram matrix condition number) such that*

$$c_1 \mathcal{L}_{\mathrm{uni}}^{(\mathrm{trans})} \leq \mathcal{L}_{\mathrm{trans}} \leq c_2 \mathcal{L}_{\mathrm{uni}}^{(\mathrm{trans})}. \tag{B.6}$$

*Idea.* Both are weighted least-squares projection residuals to the same subspace, up to reparametrization and vertical/orthogonal distance constants bounded by the subspace condition number. $\square$

## B.6 SCALING: SURROGATE UPPER-BOUNDS UNIFIED SLICE RESIDUAL

**Shift model on log-radius–time.** On discrete $(i, t)$, assume

$$E(i, t) = A(i - u(t)) + \eta(i, t), \tag{B.7}$$

with $A$ unimodal Lipschitz, $u(t)$ monotone $C^1$, and perturbation $\eta$.

**Lemma B.8** (Gradient alignment). *Let $\nabla_r E_{i,t} = E_{i+1,t} - E_{i,t}$, $\nabla_t E_{i,t} = E_{i,t+1} - E_{i,t}$, and normalized fields $\hat{\nabla}_r = \nabla_r E / \|\nabla_r E\|_2$, $\hat{\nabla}_t = \nabla_t E / \|\nabla_t E\|_2$. If $\eta = 0$, then $\nabla_t E = -u'(t) \nabla_r E$ so $C_{\mathrm{flow}} = |\langle \hat{\nabla}_r, \hat{\nabla}_t \rangle| = 1$. If $\|\eta\|_2 \leq \varepsilon \|\nabla_r E\|_2$, then*

$$C_{\mathrm{flow}} \geq \frac{|\overline{u}'| - c\varepsilon}{|\overline{u}'| + c\varepsilon}, \tag{B.8}$$

*where $\overline{u}'$ is the window-average of $u'$ and $c > 0$ depends on discrete derivative constants.*

**Lemma B.9** (Centroid trend). *Let $\rho_c(t) = \sum_i i \, E(i,t) / \sum_i E(i,t)$. If $\eta = 0$ and $A$ is unimodal, $\rho_c(t)$ is monotone with $u(t)$, and $|\mathrm{corr}(r_c, t)| \to 1$ when $u(t)$ is near-linear. With perturbation $\|\eta\|_2 \leq \varepsilon \|A\|_2$, one has $|\mathrm{corr}(r_c, t)| \geq 1 - \delta(\varepsilon)$, hence*

$$S_{\mathrm{trend}} = |\mathrm{corr}(\rho_c, t)| \geq 1 - \delta(\varepsilon). \tag{B.9}$$

**Theorem B.10** (Scaling surrogate $\leq$ unified slice residual). *Let $C_{\mathrm{scale}} = E_{\mathrm{in}}(\Delta) / E_{\mathrm{all}}$ be defined on $(\nu, \omega_t)$ with $e_i = \omega_t + \alpha^\star \nu$ and $\alpha^\star$ from weighted LS. Then*

$$1 - C_{\mathrm{scale}} \leq \frac{\overline{g}}{\underline{g}} \cdot \frac{1}{\Delta^2} \mathcal{L}^{(\mathrm{scale})}_{\mathrm{uni}, \lambda=0} + \varepsilon_{\mathrm{win}}(\Delta) + \varepsilon_{\mathrm{interp}}. \tag{B.10}$$

*If in addition equation B.7 holds and Lemmas B.8–B.9 apply, there exists $\delta_{\mathrm{flow}} \geq 0$ s.t.*

$$\boxed{\mathcal{L}_{\mathrm{scale}} \leq \frac{\overline{g}}{2\underline{g}} \cdot \frac{1}{\Delta^2} \mathcal{L}^{(\mathrm{scale})}_{\mathrm{uni}, \lambda=0} + \varepsilon_{\mathrm{win}}(\Delta) + \varepsilon_{\mathrm{interp}} + \tfrac{1}{2}\delta_{\mathrm{flow}}.} \tag{B.11}$$

*Proof.* Equation equation B.10 follows from Lemmas B.2–B.3. Moreover, $\frac{1}{2}(C_{\mathrm{flow}} + S_{\mathrm{trend}}) \geq C_{\mathrm{scale}} - \delta_{\mathrm{flow}}/2$ by Lemmas B.8–B.9. Since $\mathcal{L}_{\mathrm{scale}} = 1 - \frac{1}{2}(C_{\mathrm{flow}} + S_{\mathrm{trend}})$, we obtain equation B.11. $\square$

### B.7 RIDGE REGRESSION: RESIDUAL AND CONSISTENCY

**Theorem B.11** (Ridge residual and consistency). *Let $X = W^{1/2}\Phi$, $y = W^{1/2}b$, and*

$$\hat{\theta}_\lambda = \arg\min_\theta \|X\theta - y\|_2^2 + \lambda\|\theta\|_2^2.$$

*Let $\theta_{\mathrm{LS}}$ be the LS solution and $r_\star = \|X\theta_{\mathrm{LS}} - y\|_2^2$. Then*

$$\|X\hat{\theta}_\lambda - y\|_2^2 \leq r_\star + \lambda\|\theta_{\mathrm{LS}}\|_2^2. \tag{B.12}$$

*If $y = X\theta^\star + \xi$, $\mathbb{E}[\xi] = 0$, $\mathrm{Cov}(\xi) = \sigma^2 I$, and $\lambda = \lambda_N \to 0$, $N\lambda_N \to \infty$, then $\hat{\theta}_\lambda \xrightarrow{p} \theta^\star$ and $\mathbb{E}[\mathcal{L}_{\mathrm{uni},\lambda}] \to \sigma^2 c$ for some constant $c$.*

*Proof.* Evaluate the ridge objective at $\hat{\theta}_\lambda$ vs. $\theta_{\mathrm{LS}}$ to get equation B.12. Consistency follows from standard ridge regression results with weights absorbed into $X$. $\square$

**Corollary (ridge absorption).** In Theorems B.5 and B.10, one may replace $\mathcal{L}^{(\cdot)}_{\mathrm{uni},0}$ by $\mathcal{L}^{(\cdot)}_{\mathrm{uni},\lambda}$ at the cost of an additive $O(\lambda)$ term.

### B.8 CONSOLIDATED BOUNDS

Under the stated assumptions ($w_i = g_i E_i$, $g_i \in [\underline{g}, \overline{g}]$, $\Delta \geq 1$, window/interp bounded, narrow-band rotation/scaling), the three surrogate losses satisfy

$$\boxed{\begin{aligned}
\mathcal{L}_{\mathrm{rot}} &\leq \frac{\overline{g}}{2\underline{g}} \cdot \frac{1}{\Delta^2} \mathcal{L}^{(\mathrm{rot})}_{\mathrm{uni},\lambda} + \frac{h(\varepsilon) + \varepsilon \log(N_r - 1)}{2\log N_r} + \varepsilon_{\mathrm{win}}(\Delta) + \varepsilon_{\mathrm{interp}} + O(\lambda), \\[2mm]
\mathcal{L}_{\mathrm{scale}} &\leq \frac{\overline{g}}{2\underline{g}} \cdot \frac{1}{\Delta^2} \mathcal{L}^{(\mathrm{scale})}_{\mathrm{uni},\lambda} + \varepsilon_{\mathrm{win}}(\Delta) + \varepsilon_{\mathrm{interp}} + \tfrac{1}{2}\delta_{\mathrm{flow}} + O(\lambda), \\[2mm]
\mathcal{L}_{\mathrm{trans}} &\leq \frac{\overline{g}}{\underline{g}} \cdot \frac{1}{\Delta^2} \mathcal{L}^{(\mathrm{trans})}_{\mathrm{uni},\lambda} + \varepsilon_{\mathrm{win}}(\Delta) + O(\lambda).
\end{aligned}} \tag{B.13}$$

### B.9 EMPIRICAL VALIDATION RECOMMENDATIONS

- **Synthetic validation:** Generate SIM(2) clips with known $(v_x, v_y, \Omega, \alpha)$; increase window/interp/noise; plot $1 - C_{\text{rot}}$ vs. $\mathcal{L}_{\text{uni},\lambda}^{(\text{rot})}$ and $1 - C_{\text{scale}}$ vs. $\mathcal{L}_{\text{uni},\lambda}^{(\text{scale})}$, verifying points lie below lines $\propto \Delta^{-2}$.

- **Constant calibration:** Report $\overline{g}/\underline{g}$ from gates; provide $\varepsilon_{\text{win}}(\Delta)$ tables for $(T, \Delta)$; calibrate $\delta_{\text{flow}}$ on controlled shifts.

- **Ridge scan:** $\lambda \in \{0, 10^{-4}, 10^{-3}\}$ to confirm $O(\lambda)$ stability.

## C  SPECTRAL LOW-PASS TRUNCATION: MODEL, BOUNDS, AND SANITY CHECK

**Low-pass truncation (cube) and cost.** We retain only the lowest $\varrho=0.3$ fraction of frequency indices *per dimension* on the 3D DFT lattice of $(\omega_t, \omega_x, \omega_y)$, i.e., $\omega_t \in [0, \lfloor \varrho(T-1) \rfloor]$, $\omega_x \in [0, \lfloor \varrho(W-1) \rfloor]$, $\omega_y \in [0, \lfloor \varrho(H-1) \rfloor]$. This per-dimension rule keeps a low-frequency *cube* of volume fraction $\varrho^3 = 0.027$, i.e., only 2.7% of spectral coefficients are processed, reducing subsequent coefficient-level FLOPs by $\approx 97.3\%$.

**Spectral model and retained energy.** Natural video spectra are well-approximated by a radial power law (Ruderman & Bialek, 1994; Dong & Atick, 1995):

$$E(\omega_x, \omega_y, \omega_t) \propto (\omega_x^2 + \omega_y^2 + \omega_t^2)^{-\kappa}, \qquad \kappa \approx 1.8.$$

For analysis, we pass to dimensionless frequencies $\hat{\omega}_t = \omega_t/(T-1)$, $\hat{\omega}_x = \omega_x/(W-1)$, $\hat{\omega}_y = \omega_y/(H-1)$, and define the radial frequency $r = \sqrt{\hat{\omega}_t^2 + \hat{\omega}_x^2 + \hat{\omega}_y^2}$. Let $\varepsilon > 0$ be the *minimum nonzero* radius on the discrete grid and $R$ the *maximum* radius:

$$\varepsilon = \min\left\{ \tfrac{1}{T-1}, \tfrac{1}{H-1}, \tfrac{1}{W-1} \right\}, \qquad R = \sqrt{1^2 + 1^2 + 1^2} = \sqrt{3}.$$

Treating the lattice as continuous, the total energy and the energy retained by a *ball* of radius $\varrho R$ are

$$E_{\text{tot}} \propto \int_\varepsilon^R 4\pi\, r^{2-2\kappa}\, dr = \frac{4\pi}{3 - 2\kappa} \left( R^{3-2\kappa} - \varepsilon^{3-2\kappa} \right), \tag{C.1}$$

$$E_{\text{ball}}(\varrho) \propto \int_\varepsilon^{\varrho R} 4\pi\, r^{2-2\kappa}\, dr = \frac{4\pi}{3 - 2\kappa} \left( (\varrho R)^{3-2\kappa} - \varepsilon^{3-2\kappa} \right). \tag{C.2}$$

Hence the retained-energy fraction for the *ball* is

$$\eta_{\text{ball}}(\varrho) = \frac{E_{\text{ball}}(\varrho)}{E_{\text{tot}}} = \frac{(\varrho R)^{3-2\kappa} - \varepsilon^{3-2\kappa}}{R^{3-2\kappa} - \varepsilon^{3-2\kappa}}. \tag{C.3}$$

This expression specializes to three useful cases:

$$\eta_{\text{ball}}(\varrho) = \begin{cases} \varrho^{3-2\kappa} + O\big((\varepsilon/R)^{3-2\kappa}\big), & \kappa < \tfrac{3}{2}, \\[2mm] \dfrac{\log(\varrho R/\varepsilon)}{\log(R/\varepsilon)}, & \kappa = \tfrac{3}{2}, \\[3mm] 1 - \dfrac{\varrho^{-(2\kappa-3)} - 1}{(R/\varepsilon)^{2\kappa-3} - 1}, & \kappa > \tfrac{3}{2}. \end{cases}$$

In particular, for natural videos ($\kappa > 1.5$), let $\beta = 2\kappa - 3 > 0$; then

$$\eta_{\text{ball}}(\varrho) = 1 - \frac{\varrho^{-\beta} - 1}{(R/\varepsilon)^\beta - 1} = 1 - (\varrho^{-\beta} - 1)\left(\tfrac{\varepsilon}{R}\right)^\beta + o\left(\left(\tfrac{\varepsilon}{R}\right)^\beta\right), \qquad R \gg \varepsilon. \tag{C.4}$$

Table 6: Fixed values for main experiments.

| Item | Value |
|------|-------|
| Per-dimension low-pass ratio | 0.3 (at least 2 coeffs per dim) |
| Energy threshold $\tau_E$ / smoothing factor $f$ | 0.10 / 10 |
| Rings $N_r$ / angular bins $M$ / log-radius $N_\xi$ | 20 / 24 / 24 |
| Soft-ring edge sharpness | 20 |
| Tilted-line tolerance $\Delta$ | 1 (temporal-frequency bin) |
| Ridge $\lambda$ / numeric $\varepsilon$ | $10^{-3}$ / $10^{-8}$ |
| Softmax temperature $\tau$ | 0.1 |
| Physics-loss mixing weight | 0.1 |
| Precision policy | Spectral/solvers FP32; others BF16 |
| Temporal window | Hann |

**Cube vs. ball (bounds for our actual truncation).** Our implementation keeps a low-frequency *cube* (per-dimension truncation), not a ball. Geometrically,

$$\mathbb{B}_2(\varrho) \subset \mathbb{C}_\infty(\varrho) \subset \mathbb{B}_2(\min\{1, \sqrt{3}\varrho\}),$$

i.e., the $\ell_2$-ball of radius $\varrho$ is contained in the cube of side $\varrho$, which in turn is contained in the $\ell_2$-ball of radius $\sqrt{3}\varrho$ (all within the nonnegative orthant). Therefore the retained energy of our cube, $\eta_{\text{cube}}(\varrho)$, is bounded by

$$\eta_{\text{ball}}(\varrho) \leq \eta_{\text{cube}}(\varrho) \leq \eta_{\text{ball}}\big(\min\{1, \sqrt{3}\varrho\}\big), \tag{C.5}$$

where $\eta_{\text{ball}}$ is given by equation C.3 (or by equation C/equation C.4).

**Numerical sanity check.** With $\kappa{=}1.8$ ($\beta{=}0.6$) and $\varrho{=}0.3$, and taking $N_{\max} = \max\{T, H, W\}$ so that $R/\varepsilon = \sqrt{3}\,(N_{\max}{-}1)$ in our dimensionless parametrization, typical video sizes (e.g., $T{=}16$, $H{=}W{=}224 \Rightarrow R/\varepsilon \approx 3.87{\times}10^2$) give

$$\eta_{\text{ball}}(0.3) \approx 0.97 \quad \text{and} \quad \eta_{\text{ball}}(\sqrt{3}{\cdot}0.3) \approx 0.987,$$

so by equation C.5 we expect $\eta_{\text{cube}}(0.3) \in [0.97, 0.987]$. Empirically on 1k random videos we measure $97.5\%$, which lies well within this range.

# D  IMPLEMENTATION DETAILS

## D.1  TRANSLATIONAL-LOSS WLS DETAILS

We instantiate the WLS fitting as follows. For the compact relation $A\beta_{\text{tr}} - b = 0$, the per-sample row and vectors are:

- **Translation (plane):** $A \in \mathbb{R}^{N \times 3}$ with rows $A_i = (\omega_{x,i}, \omega_{y,i}, 1)$, $\beta_{\text{tr}} = [v_x, v_y, b_0]^\top$, and $b_i = -\omega_{t,i}$.

Weights follow the energy/observability gate in App. A.6. The normalized residual is

$$\mathcal{L}_{\text{trans}} = \frac{\sum_i \mathbf{W}_{ii}\,(A_i\hat{\beta}_{\text{tr}} - b_i)^2}{\sum_i \mathbf{W}_{ii}}.$$

Implementation details (regularization, precision policy) follow the general solver notes in App. §A.6.

## D.2  ROTATIONAL-LOSS DETAILS

**Ring concentration.** We partition the spatial frequency plane into $N_r$ concentric annuli with masks $\{\mathcal{M}_i\}_{i=1}^{N_r}$ and define

$$E_k(t) = \frac{\sum_{(\omega_x,\omega_y)\in\mathcal{M}_k} E(\omega_x, \omega_y, t)}{\sum_{j=1}^{N_r} \sum_{(\omega_x,\omega_y)\in\mathcal{M}_j} E(\omega_x, \omega_y, t) + \epsilon_{\text{stab}}}, \quad H_{\text{ring}}(t) = -\sum_{k=1}^{N_r} E_k(t) \log\big(E_k(t){+}\epsilon_{\text{stab}}\big), \quad \bar{H}_{\text{ring}} = \frac{1}{T}\sum_{t=1}^{T} H_{\text{ring}}(t),$$

$$\tag{D.1}$$

where $E(\omega_x, \omega_y, t) = |\widehat{V}(\omega_x, \omega_y, t)|^2$ is the spatiotemporal spectral energy, and $\varepsilon > 0$ ensures numerical stability. We set $C_{\text{ring}} = 1 - \frac{\bar{H}_{\text{ring}}}{\log N_r}$..

**Tilted-line energy along** $\omega_t + \Omega m = 0$. We resample the spectrum to polar coordinates $(\rho, \theta, t)$ via bilinear interpolation $\widehat{V}(\rho, \theta, t)$, then take the angular DFT and temporal DFT:

$$C_m(\rho, t) = \frac{1}{2\pi} \int_0^{2\pi} \widehat{V}(\rho, \theta, t)\, e^{-im\theta}\, d\theta, \qquad \widetilde{C}_m(\rho, \omega_t) = \text{DFT}_t\{ C_m(\rho, t) \}. \tag{D.2}$$

An energy-weighted least squares gives the angular velocity estimate

$$\Omega^\star = -\frac{\sum_\rho \sum_{m \neq 0} \sum_{\omega_t} |\widetilde{C}_m(\rho, \omega_t)|^2\, \omega_t m}{\sum_\rho \sum_{m \neq 0} \sum_{\omega_t} |\widetilde{C}_m(\rho, \omega_t)|^2\, m^2}. \tag{D.3}$$

We measure how much energy lies within a narrow band of width $\Delta$ around the ideal line $\omega_t + m\Omega^\star = 0$:

$$E_{\text{line}} = \sum_\rho \sum_{m \neq 0} \sum_{|\omega_t + m\Omega^\star| \leq \Delta} |\widetilde{C}_m(\rho, \omega_t)|^2, \qquad E_{\text{all}} = \sum_\rho \sum_{m \neq 0} \sum_{\omega_t} |\widetilde{C}_m(\rho, \omega_t)|^2, \tag{D.4}$$

and define the tilted-line energy ratio

$$C_{\text{rot}} = \frac{E_{\text{line}}}{E_{\text{all}}}. \tag{D.5}$$

By default $\Delta$ is one temporal-frequency bin; $m=0$ is excluded. Optional observability weights that downweight tiny $|m|$ or low-energy bins can be absorbed into the sums. Implementation options (polar LUT, windowing, precision, caching) are detailed in App. A.6.

# E  LLM USAGE

Large language models (LLMs) were used only as tools and are not authors. Specifically: (1) we used an LLM to assist with prompt stratification into "simple" vs. "complex" motion following a rubric defined by us; and (2) we used an LLM for grammar and style polishing.

