# OpenReview forum: "Physics-Guided Motion Loss for Video Generation Model"
_ICLR.cc/2026/Conference — ICLR 2026 Conference Desk Rejected Submission_

### Official Review · Reviewer_2TB5 · 2025-10-26

**Soundness:** 3
**Presentation:** 3
**Contribution:** 2
**Rating:** 6
**Confidence:** 2

**Summary:**

This paper introduces a frequency-domain physics-guided framework to improve the physically plausible motion in video diffusion. This paper formulates rigid motions (translation, rotation, and scaling) within a unified spectral SIM(2) framework, and proposes corresponding differentiable frequency-domain losses. Empirically, the proposed method improves motion accuracy and temporal coherence across multiple backbones (Open-Sora, MVDIT, Hunyuan).

**Strengths:**

- This paper explores an important problem in video generation.
- The SIM(2)-based spectral derivation unifies translation, rotation, and scaling within a mathematically sound framework.
- The loss is architecture-agnostic and can be inserted into any diffusion model without modifying the backbone.
- Evaluation spans three major video diffusion systems and includes multiple metrics.

**Weaknesses:**

- The innovation lies mainly in unifying them under the SIM(2) formulation.
- The method only addresses translation, rotation, and scaling, which limits applicability to real-world complex scenes.
- There are some related physics-constrained video generation works, such as [a], which should also be discussed. Also, except for comparing with the baseline models, it should compare with some related works.
- [b] is a comprehensive physics generation benchmark designed to evaluate physical commonsense correctness in T2V generation. To validate the effectiveness of the proposed method, it is suggested to apply [b].
- Since the proposed method is plugged into the baseline models, it should report the generation times.
- How to set the temperature parameter τ? And how sensitive it is to τ?


[a] MOTIONCRAFT: Physics-based Zero-Shot Video Generation
[b] Towards World Simulator: Crafting Physical Commonsense-Based Benchmark for Video Generation

**Questions:**

See the detailed comments in weaknesses.

---

> ### Author Response · Authors · 2025-11-21
>
> We thank the reviewer for the careful reading and constructive feedback. We address the questions below.
>
> Q1.Temperature τ.
> We use τ=0.1 by default (Table 6). The temperature parameter τ controls the sharpness of softmax‑based mixing: small τ yields winner‑takes‑all; larger τ averages across motion types.
>
> Q2.Generation time.
> Our method does not modify the backbone or sampler; inference compute remains unchanged by design (A.7).
>
> Q3.Relation to MotionCraft and physics benchmarks.
> We have added the discussion of MotionCraft in the revised paper "MotionCraft imposes physics-based motion at inference via simulated flow warping of image-diffusion noise, while our method regularizes motion during video-diffusion training in the frequency domain". Following the reviewer’s suggestion, we also evaluate on the recent Physics Generation Benchmark: our method improves the average physics score of Open‑Sora from 0.44 to 0.52 (Table 4), indicating better adherence to basic physical laws.
>
> W1.Only translation/rotation/scaling.
> We focus on the most common global rigid motions (translation / rotation / scaling) and use them as an architecture‑agnostic prior. Two observations mitigate the concern: (i) on a complex‑motion subset, we still observe consistent gains (Table 3); (ii) the softmax mixing emphasizes whichever motion patterns best match a window without hard classification.
>
> W2.Novelty.
> Our goal is to make the SIM(2) frequency geometry usable as a simple, backbone-agnostic training prior for video diffusion. We (i) turn the classical SIM(2) hyperplane into three lightweight, differentiable slice losses with theoretical upper-bound relations to a unified spectral residual (App. A–B), (ii) combine them with low-pass 3D FFT and adaptive softmax mixing that can be dropped into existing backbones without architectural changes, and (iii) show consistent motion and physics gains across three video diffusion systems.

---

> > ### Comment · Reviewer_2TB5 · 2025-11-26
> >
> > My concerns have been addressed, and I would like to maintain my positive rating.

---

### Official Review · Reviewer_ffbp · 2025-10-31

**Soundness:** 3
**Presentation:** 3
**Contribution:** 3
**Rating:** 8
**Confidence:** 3

**Summary:**

This paper proposes a physics guided motion loss for video generation models, designed to enhance motion plausibility and easily integrate with any existing video diffusion models. The loss regularizes fundamental physical motions (rotation, translation, and scaling) in the frequency domain, where these motions exhibit simple and easily detectable patterns. Both qualitative video results and quantitative evaluations demonstrate the effectiveness of the proposed loss.

**Strengths:**

It is novel and interesting to regularize basic global physical motions in the frequency domain, a simple yet effective approach that can be easily integrated into any video generation model.

**Weaknesses:**

1. The applicable physics motion patterns are limited to rotation, translation and scaling.
2. My understanding is that the method is primarily effective for videos containing a single dominant motion and cannot handle scenarios involving multiple objects moving differently or simultaneous camera and object motion.

**Questions:**

1. If a video contains both camera motion and object motion, for example, when the camera is rotating or zooming while an object translates across the scene, can the proposed method still capture both types of motion?

2. I am also curious how the method performs on static scenes with only camera motion compared to models that explicitly use camera motion as control signals.

---

> ### Author Response · Authors · 2025-11-21
>
> We thank the reviewer for the careful reading and constructive feedback. We address the questions below.
>
>
> Q1.Co‑occurring camera and object motion.
> The three losses are computed on distinct spectral coordinates and mixed via a softmax weighting. They are not mutually exclusive: e.g., a rotating/zooming camera excites rotational/scaling slices, while an object translation excites the translation plane; all relevant slice losses can be active simultaneously during training (please check Fig. 2—new result). We see corresponding gains on prompts with mixed motions (Table 3).
>
> Q2.Camera‑only motion.
> A static scene observed by a moving camera is well‑approximated by a SIM(2) transform over short windows. Relation to camera‑control methods: Approaches that ingest trajectories/extrinsics can enforce a prescribed path at inference time but require additional control inputs; our training‑only spectral regularizer is complementary and leaves inference compute unchanged.
>
> W1. Multiple moving objects.
> By linearity of the 3D FFT, a multi‑body scene yields an approximately additive superposition of each body’s spectral pattern (please check new result figure 2); our energy‑weighted fitting then prioritizes the dominant components in each window, while others contribute to the residual. Consequently, our method remains compatible with scenes containing multiple moving objects.

---

### Official Review · Reviewer_1ggH · 2025-11-03

**Soundness:** 4
**Presentation:** 2
**Contribution:** 4
**Rating:** 8
**Confidence:** 4

**Summary:**

This paper introduces a frequency-based, physics-informed approach to enhance motion quality and physical plausibility in video diffusion models without degrading visual fidelity or text alignment.
Rather than operating purely in pixel or latent space, the method injects a low-pass frequency-domain constraint that regularizes temporal dynamics and enforces physically consistent motion patterns.
The approach is applied as an auxiliary frequency-domain loss during training, compatible with existing architectures such as Open-Sora, MVDIT, and Hunyuan , by training a LoRA. It requires no modification to the diffusion backbone and adds only moderate computational cost.
Main contributions are:
-  A physics-guided frequency-domain regularization for video diffusion training that improves temporal motion realism.
-  Efficient low-pass truncation scheme reducing computational cost. Differentiable frequency-domain least-squares loss integrated seamlessly into standard diffusion training loops.
-  Extensive empirical validation on multiple video diffusion models.

**Strengths:**

The paper introduces a novel frequency-domain regularization for video diffusion models that leverages spectral signatures of translation, rotation, and scaling to guide learning without altering model architecture. The strengths are:
-  The idea of combining classical ideas from Fourier analysis and the SIM(2) motion group with modern video diffusion models, demonstrating a creative synthesis of physics-based priors and deep generative modeling.
-  The authors provide a thorough derivation connecting basic physical motions (translation, rotation, scaling) to spectral signatures, with attention to windowing, interpolation errors, and numerical stability. The breakdown of translational, rotational, and scaling motion losses, along with adaptive weighting, is logically organized and explained with intuitive interpretations.
-  Results are reported on multiple video diffusion backbones and evaluated on diverse metrics. The experiments are comprehensive (no LoRA, +LoRA with other losses, +LoRA with proposed loss), and quantitative gains are consistent.

**Weaknesses:**

-  Although the theory is solid, as a paper in the video generation field, its presentation lacks some intuitive visualizations, such as visual demonstrations of spectral changes, and the qualitative evaluation is relatively limited;
-  In the Abstract, “regularizer” is written as “regular- izer,” which looks like a copy-paste error;
-  On the first page, in the “four groups” listing, why only (i) is bolded;
-  As an important demonstration, the supplementary video is of poor asthetic quality and needs improvement.

Overall, the paper has no significant issues in theory or experiments, but there are some minor presentation problems.

**Questions:**

-  Does this method support multiple object motions? (e.g. if a single apple is cut in half and the two halves split away.)
-  Does this method support the color, illumination, or texture change of an object within a video? If so, how does this loss reduce color flickering like the train in Appendix Fig. 4 ?

---

> ### Author Response · Authors · 2025-11-21
>
> We thank the reviewer for the careful reading and constructive feedback. We address the questions below.
>
> Q1.Does this method support multiple object motions?
> Our method is computed on short-window global spatio‑temporal spectra, so it remains well‑defined when several rigid bodies move simultaneously(please check Figure 2—new result). By linearity of the 3D FFT, a multi‑body scene yields an approximately additive superposition of each body’s spectral pattern; our energy‑weighted fitting then prioritizes the dominant components in each window, while others contribute to the residual (a soft prior rather than a hard constraint). The improvements persist on a “complex‑motion” subset (Table 3), indicating robustness to mixed motions.
>
> Q2. Does the method support appearance changes and how does the loss reduce color flickering?
> Our method operates on frequency‑domain energy geometry, so it does not assume strict brightness constancy. Moderate color/illumination/texture changes preserve the SIM(2) structure of motion slices, while irregular flicker injects transient energy without SIM(2) geometry and is penalized in the residual. We observe consistent reductions in Warping Error and improvements in Temporal Consistency across backbones (Tables 1–2), aligning with the reduced flicker example.
>
> W1. Visual demonstrations of spectral changes,  video asthetic quality.
> We added Figure 2 in the revised version to provide intuitive visualizations of the spectral changes. We also improved the aesthetic quality of the video, and fixed the typos.

---

### Author Response · Authors · 2025-12-04
**Revision Summary**

We thank all reviewers for their constructive feedback. Below is a summary of key revisions:

- Theory and Visualization (Sec. 3.1, Fig. 2): We added spectral visualizations of SIM(2) signatures and clarified how our method handles multiple moving objects and co-occurring camera/object motion via FFT linearity, addressing Reviewers 1ggH and ffbp.
- Related Work and Benchmarks (Sec. 2, Table 4): Following Reviewer 2TB5's suggestions, we added a MotionCraft discussion and evaluated on the Physics Generation Benchmark, showing consistent improvements over the baseline.
- Implementation Details (Table 6, App. A.7): Clarified that τ=0.1 by default and that inference cost remains unchanged.
- Presentation: Fixed typos and improved supplementary video quality per Reviewer 1ggH.

Reviewer 2TB5 has confirmed that concerns have been addressed. We hope this summary assists in the final assessment of our work.

---

### Note · Program_Chairs · 2026-01-17
**Submission Desk Rejected by Program Chairs**

The following references in this submission do not refer to real documents and/or have major errors in bibliographic information:

 Jun Chen, Peter Kellman, and Gonzalo R Arce. Scale-space theory for the frequency domain: Properties and applications of the frequency-domain scaling transform. IEEE Transactions on Image Processing, 19(11):2798-2810, 2010.